# MACROGUIDE: Topological Guidance for Macrocycle Generation

Alicja Maksymiuk [* 1 2]   Alexandre Duplessis [* 3]   Michael Bronstein [1 2]   Alexander Tong [2]   Fernanda Duarte [1]
İsmail İlkan Ceylan [4 2 1]

## Abstract

Macrocycles are ring-shaped molecules that offer a promising alternative to small-molecule drugs due to their enhanced selectivity and binding affinity against difficult targets. Despite their chemical value, they remain underexplored in generative modeling, likely owing to their scarcity in public datasets and the challenges of enforcing topological constraints in standard deep generative models. We introduce MACROGUIDE: Topological Guidance for Macrocycle Generation, a diffusion guidance mechanism that uses Persistent Homology to steer the sampling of pretrained molecular generative models toward the generation of macrocycles, in both unconditional and conditional (protein pocket) settings. At each denoising step, MACROGUIDE constructs a Vietoris-Rips complex from atomic positions and promotes ring formation by optimizing persistent homology features. Empirically, applying MACROGUIDE to pretrained diffusion models increases macrocycle generation rates from 1% to 99%, while matching or exceeding state-of-the-art performance on key quality metrics such as chemical validity, diversity, and PoseBusters checks. The implementation can be found at https://github.com/ala1705/MacroGuide.

## 1. Introduction

Macrocycles – cyclic molecules with a ring of 12 or more heavy atoms – have attracted growing interest as drug candidates due to their improved *selectivity* and *binding affinity* against difficult targets (Garcia Jimenez et al., 2023; Mallinson & Collins, 2012; Giordanetto & Kihlberg, 2014). The improvement in selectivity is largely attributed to the ring structure, which restricts molecular flexibility relative

*Equal contribution [1]University of Oxford [2]AITHYRA [3]ENS Ulm, PSL [4]TU Wien. Correspondence to: Alicja Maksymiuk <alicja.maksymiuk@cs.ox.ac.uk>.

*Proceedings of the 43rd International Conference on Machine Learning*, Seoul, South Korea. PMLR 306, 2026. Copyright 2026 by the author(s).

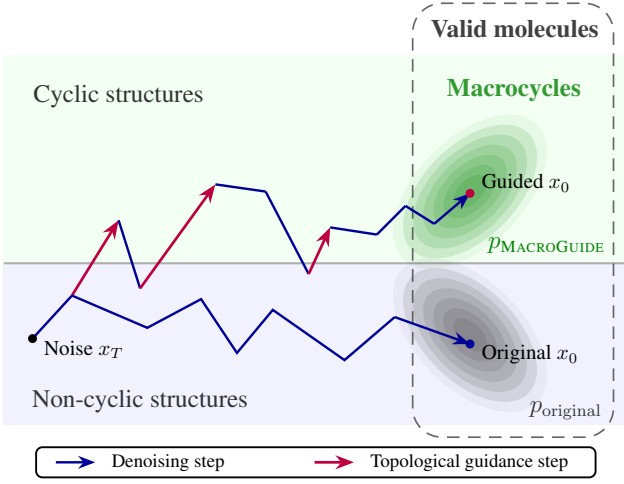

*Figure 1.* **Method overview.** MACROGUIDE drives the denoising trajectory towards macrocyclic structures using updates from a topological objective.

to linear analogs. This reduces off-target binding arising from conformational changes. Furthermore, macrocycles' larger size enables more protein-ligand interactions, allowing them to target challenging binding sites or even external protein surfaces (Yudin, 2015). The improvement in binding affinity comes from macrocycles' greater rigidity too, as it decreases their entropy in solvent and hence reduces entropic penalty upon binding.

**Macrocycles in drug discovery.** Many macrocycles can fold into conformations that mask polar groups (Naylor et al., 2017), which allows them to achieve oral bioavailability and to defy classical drug-likeness rules (e.g. Rule of 5 by Lipinski et al. (1997)). Macrocycles play a critical role in modern therapeutics with 17 macrocyclic drugs approved by the FDA in the last five years alone (Du et al., 2025) (out of over 75 approved to date (Jiang et al., 2026)). This includes clinical successes such as the immunosuppressant cyclosporine or the oncological drug lorlatinib (which outperforms its linear counterpart, crizotinib (Ermert, 2017; Shaw et al., 2020)). Despite these developments and the great therapeutic promise of macrocycles, they remain underexplored in deep generative modeling.

**Macrocycle generation.** Existing work on macrocycle[1] generation has mostly focused on cyclic peptides (see Appendix A.1 for a literature overview). These methods leverage peptide-specific traits, such as backbone homogeneity arising from a small set of amino-acid building blocks, which makes them inapplicable to structurally-diverse, arbitrary macrocycles.

Non-peptidic macrocycle design has been explored in two prior works. Macformer, a SMILES-based transformer trained on linear-to-macrocyclic pairs, exploits the macrocyclization of pre-existing linear scaffolds to generate JAK2 inhibitors (Diao et al., 2023). Macro-Hop performs macrocycle scaffold hopping via reinforcement learning and successfully produces PDGFR$\alpha$ inhibitors by generating structures that satisfy predefined constraints and 3D similarity to a reference molecule (Liang et al., 2025). However, both methods rely on handcrafted scaffolds or linear precursors, which severely limits their applicability in *de novo* design, where neither the appropriate reference nor a linear equivalent is known *a priori*.

**Problem statement and challenges.** Despite the rapid progress of diffusion-based models for small-molecule design (Wang et al., 2025; Peng et al., 2023; Hoogeboom et al., 2022; Vignac et al., 2023a), to the best of our knowledge, *there is currently no generative method explicitly designed to produce arbitrary macrocycles*. This gap can be largely attributed to two factors. First, public chemical datasets contain relatively few macrocycles, often much simpler than therapeutically-relevant ones (Garcia Jimenez et al., 2023). Second, the existence of a large ring is a global topological property, whereas most generative models focus on approximating local chemical validity. As a result, existing unconstrained generative algorithms rarely produce macrocycles (see Table 1).

**Contributions.** We introduce Topological Guidance for Macrocycle Generation (MacroGuide), a diffusion guidance mechanism that steers pretrained molecular generative models – originally designed for unconstrained molecule generation – towards the generation of macrocycles, with or without conditioning (Figures 1 and 2). It works by computing gradients of the persistent homology features of a Vietoris-Rips complex at each step of the denoising process.

- Conceptually, MacroGuide is (i) *training-free*, generating macrocyclic topologies that may be rare or completely absent from the training data of the base model, without a need for retraining or finetuning; (ii) *lightweight*, introducing minimal computational overhead; (iii) *general*, as it can be plugged into various diffusion-based models; and

---

[1]In this paper, *macrocycles* refer to molecules with a cycle of at least 12 heavy atoms, although this term is sometimes used to describe a specific subset of such molecules, namely *cyclic peptides*, which are built out of amino acids and peptide bonds.

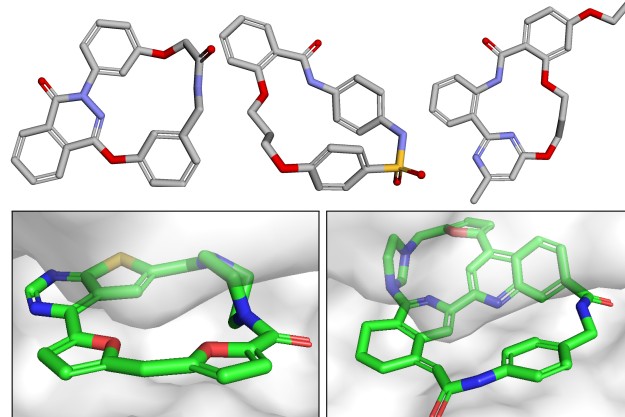

*Figure 2.* **Examples of generated macrocycles.** *Top:* Unconditional generation. *Bottom:* Protein conditioning. *Bottom right:* This molecule was specifically optimized to be bicyclic (two rings). Molecule fragments appear transparent when hidden by parts of the protein pocket.

(iv) *flexible*, allowing the user to specify both the number of rings and their sizes.

- Experimentally, we evaluate MacroGuide in both unconditional and conditional (protein pocket) settings; demonstrating a 99% rate of macrocycle generation, compared to a baseline of 0%-5%. Importantly, our method matches or exceeds state-of-the-art models across key quality metrics such as chemical validity, diversity, PoseBusters checks (Buttenschoen et al., 2024), and pharmacophore satisfaction.

This work introduces *the first de novo arbitrary macrocycle generation method*, addressing a critical gap in deep generative modeling for drug discovery.

## 2. Preliminaries

### 2.1. Diffusion-based Molecule Generation

Diffusion models (Sohl-Dickstein et al., 2015; Ho et al., 2020; Song & Ermon, 2019) are a class of generative models that produce data by reversing a gradual noising process. In the forward process, a clean sample $x_0$ is progressively perturbed according to a Markov chain yielding noisy samples $x_t$ that converge to an isotropic Gaussian as $t$ increases. At the core of these models is the learning of the *score function*

$$\mathbf{s}_\theta(x_t, t) \approx \nabla_{x_t} \log p_t(x_t),$$

which estimates the gradient of the log-density of the perturbed data. It is then applied iteratively, starting from pure noise, to generate samples during the reverse process, also known as denoising (Song & Ermon, 2019; Song et al., 2021).

Diffusion models have been applied to tasks such as *de novo* molecule generation (Hoogeboom et al., 2022; Vignac

et al., 2023a; Peng et al., 2023; Schneuing et al., 2024; Ziv et al., 2025; Schneuing et al., 2025), as their iterative denoising formulation allows models to capture global structural dependencies while refining fine-grained details. In the molecular setting, diffusion can either operate directly in continuous 3D coordinate space of atoms (Hoogeboom et al., 2022), on molecular graphs represented as adjacency matrices (Vignac et al., 2023a), or in a natural combination of both where the atom positions, atom types and bonds are included in the denoising network (Peng et al., 2023; Vignac et al., 2023b). In contrast to sequential (autoregressive) models, diffusion-based approaches can more easily generate molecules in their energetic minima without the need for further redocking, as they model the full joint distribution of atoms.

## 2.2. Persistent Homology for Molecular Point Clouds

Persistent homology is a tool from topological data analysis (TDA) that characterizes the topology of data across multiple scales (Edelsbrunner et al., 2002; Edelsbrunner & Harer, 2010). In the context of a molecular 3D point cloud, one constructs a family of *simplicial complexes* that encode proximity relationships between points, forming a nested sequence known as a *filtration*. A common choice is the *Vietoris-Rips complex* at scale $\varepsilon$:

$$\mathrm{VR}_\varepsilon(X) = \{\sigma \subseteq X \mid d(x_i, x_j) \leq \varepsilon; \forall x_i, x_j \in \sigma\},$$

where $d$ is typically the Euclidean distance. As $\varepsilon$ increases, the topology of $\mathrm{VR}_\varepsilon(X)$ evolves: connected components (captured by $H_0$) merge, loops and cycles ($H_1$) form and fill in, and in higher dimensions ($H_2$, etc.) voids appear and vanish. For each component $p_i^{(D)}$ of dimension $D$, persistent homology records the *birth* $b_i^{(D)}$ and *death* $d_i^{(D)}$ scales, producing a *persistence diagram* or *barcode*. In the context of molecules, $H_0$ features correspond to clusters of atoms, $H_1$ components often correspond to chemical rings or other cycle-like structures, and $H_2$ can capture internal cavities, relevant for instance in molecular cages. These descriptors are invariant to rigid transformations, robust to small perturbations and differentiable, making them useful for integrating topology information into various machine learning models (see Appendix A.2 for a literature overview).

## 3. Methodology

### 3.1. Method Overview

For each diffusion time step $t$, we define the state of a molecule as $M_t = \{X_t, A_t, B_t\}$, where $X_t \in \mathbb{R}^{N \times 3}$ describes the positions of the $N$ atoms, $A_t \in [0,1]^{N \times a}$ is the probability distribution on their atomic types (with $a$ the number of atom types), and $B_t \in [0,1]^{N \times N \times b}$ is the distribution on the covalent bond types (if available), with $b$

the number of allowed bond types, including the absence of a bond.

We update the score provided by the denoising architecture $s_\theta(X_t, t)$ with our guidance function $\mathcal{F}_{\mathrm{TDA}}$

$$\tilde{s}_\theta(X_t, t) = s_\theta(X_t, t) - \lambda_t \nabla_{X_t} \mathcal{F}_{\mathrm{TDA}}(X_t) \quad (1)$$

where $\lambda_t$ allows for scheduling of the guidance.

The parameters of the original architecture remain fixed and the network is *not* retrained. The guidance term (Figure 3) is based on the computation of a Vietoris-Rips complex, and is defined as

$$\mathcal{F}_{\mathrm{TDA}}(X) = F_{\mathrm{death}}^{H_1}(X) + F_{\mathrm{birth}}^{H_1}(X) + F_{\mathrm{death}}^{H_0}(X) \quad (2)$$

- **Cycle size – $H_1$ death.** $F_{\mathrm{death}}^{H_1}$ encourages the size of the largest cycle to be in a given range. Specifically, we optimize $d_{i^*}^{(1)}$ (where $i^\star = \arg\max_i d_i^{(1)}$), the death time of the $H_1$ component (ring) that dies last, to lie in a target interval $[d_{\min}, d_{\max}]$:

$$F_{\mathrm{death}}^{H_1}(X) = \Big(\mathrm{ReLU}\big(d_{\min} - d_{i^\star}^{(1)}(X)\big)\Big)^2 + \Big(\mathrm{ReLU}\big(d_{i^\star}^{(1)}(X) - d_{\max}\big)\Big)^2 \quad (3)$$

This promotes the formation of a large cycle and enables control over its size. The link between the number of atoms in the cycle and the death time of its associated $H_1$ component is detailed in Section 3.3.

- **Cycle connectivity – $H_1$ birth.** $F_{\mathrm{birth}}^{H_1}$ acts as a proxy for cycle connectivity, by ensuring each edge is not longer than the maximum allowed bond length $\ell^\star$. In practice

$$F_{\mathrm{birth}}^{H_1}(X) = \mathrm{ReLU}\big(b_{i^\star}^{(1)}(X) - \ell^\star\big) \quad (4)$$

constrains the largest edge size in the cycle $b_{i^\star}$. Since bond information is not explicitly modeled, $F_{\mathrm{birth}}^{H_1}$ does not formally guarantee cycle connectivity, but we find it sufficient in practice. The absence of a squared penalty for this term, in contrast to the other guidance terms, empirically leads to improved performance (see Appendix J.3).

- **Molecule connectivity – $H_0$ death.** $F_{\mathrm{death}}^{H_0}$ promotes the existence of one single connected component by making sure all the distances between any two adjacent atoms (i.e. the death $d_j^{(0)}$ of an $H_0$ component) are of length less than the maximum allowed bond length $\ell^\star$:

$$F_{\mathrm{death}}^{H_0}(X) = \sum_{j=1}^{N_0} \Big(\mathrm{ReLU}\big(d_j^{(0)}(X) - \ell^\star\big)\Big)^2 \quad (5)$$

where $N_0$ is the number of finite-death $H_0$ components.

Detailed algorithms can be found in Appendix F.

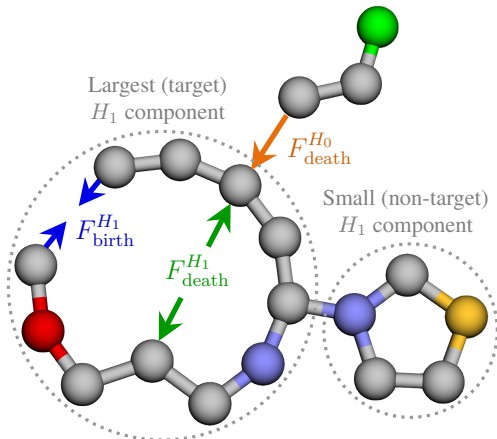

*Figure 3.* **Topological guidance for diffusion-based macrocycle generation.** Bonds were added here to convey chemical scale but are not used by the guidance, which operates only on atom coordinates.

**Addressing gradient sparsity.** Each topological feature is created and destroyed by specific simplices. As a consequence, the gradient signal comes only from the few points involved in critical birth and death events (Poulenard et al., 2018). Although this is generally not problematic, as the critical points typically change across denoising steps (Carrière et al., 2021), it can render the optimization of $H_0$ components unstable and potentially divergent. We address this issue by masking the gradient of the atom closest to the molecular centroid. Further details and theoretical guarantees are provided in Appendix D.

### 3.2. Posterior Sampling

Following Guo et al. (2024), the score of a conditional distribution admits the decomposition

$$\nabla_x \log p_t(x \mid y) = \nabla_x \log p_t(x) + \nabla_x \log p_t(y \mid x).$$

In our setting, the guidance term can thus be interpreted as a conditional score term

$$\nabla_x \log p_t(y \mid x) = -\lambda_t \, \nabla_x \mathcal{F}_{\text{TDA}}(x), \qquad \lambda > 0, \quad (6)$$

which corresponds to an energy-based conditional likelihood

$$p(y \mid x) \propto \exp\big(-\lambda_t \, \mathcal{F}_{\text{TDA}}(x)\big).$$

In particular Bayes' Theorem gives

$$p_t(x \mid y) \propto p_t(x) \exp\big(-\lambda_t \, \mathcal{F}_{\text{TDA}}(x)\big),$$

and guidance can be interpreted as approximating sampling from this conditional distribution in the continuous-time limit with an exact score.

In practice, however, this interpretation only holds approximately. First, the learned score is imperfect, so the dynamics do not exactly track the target SDE. Gao et al. (2025) analyze this discrepancy and propose rectified guidance formulations to reduce bias from score errors. Second, practical samplers discretize the SDE. Guo et al. (2024) show that discretization introduces systematic deviations in the effective distribution, and propose variance-preserving corrections. Nevertheless, we find this approximation sufficient for our task, as shown in Section 4.

### 3.3. Macrocycle Size Control

**Combinatorial and geometric sizes.** The *size* of a macrocycle is defined here as the number $n$ of heavy atoms it contains. However, we only have direct control over a geometric notion of size, which is the *death* of the corresponding $H_1$ component in the Vietoris-Rips filtration. Therefore we need to establish an explicit relationship between these two definitions.

A convenient and analytically tractable way to do so is to consider an idealized but informative model: an equilateral polygon with $n$ vertices embedded in the plane (representing atoms), and edges of length $\ell$ (representing bonds). However, regularity is not a realistic assumption as tetrahedral carbon arrangements impose smaller bond angles than the ones of a regular polygon ($\theta = 109.5°$, which can be observed in Figure 2). We thus study crown-shaped equilateral $n$-gons with interior angles $\theta$ (see Figure 8 of Appendix D for a visualization). The following theorem provides a simple relationship $n = f(d, \ell)$ in this framework (Proof in Appendix C).

**Theorem 3.1** (Vietoris-Rips death time of a tetrahedral cycle). *Consider a regular cyclic conformation of n atoms (n even) with bond length $\ell$ and bond angle $\theta$. The death time d of the dominant $H_1$ component in the Vietoris-Rips filtration is given by:*

$$d = \ell \sqrt{2(1 - \cos\theta)} \cdot \frac{\sin\left(\left\lceil \frac{n}{6} \right\rceil \frac{2\pi}{n}\right)}{\sin\left(\frac{2\pi}{n}\right)} \qquad (7)$$

*In the limit of large n,*

$$n = \frac{4\pi}{\sqrt{6(1 - \cos\theta)}} \frac{d}{\ell} + \mathcal{O}(1) \qquad (8)$$

**Accuracy of the linear approximation.** Solving Equation (7) for $n$ does not admit an easy closed-form solution, making the linear approximation of Equation 8 very attractive from a practical point of view, assuming it is tight enough. Consider a macrocycle of size $n = 16$ (crown shaped) with a typical bond length $\ell = 1.5$ Å, and $\theta = 109.5°$. The linearization provided incurs a relative error of approximately 10%. This confirms that for drug-like macrocycles, the higher-order terms are negligible in practice. We further validate Equation 8 empirically in Section 4.3.1.

# 4. Experiments

## 4.1. Main Results

### 4.1.1. UNCONDITIONAL GENERATION

To demonstrate the ability of MACROGUIDE to generate macrocycles we apply it to MolDiff, a diffusion model that generates molecules by denoising a 3D point cloud of atoms, together with bonds (Peng et al., 2023). MolDiff was pre-trained on GEOM-Drug (Axelrod & Gomez-Bombarelli, 2022), a dataset of small molecules with drug-like properties, where only 0.14% of the training data are macrocycles. In our guidance method, the simplicial complex is built using Pytorch Topological (Rieck, 2022). Further setup details are described in Appendix G.

**Baselines.** As MACROGUIDE is the first arbitrary macrocycle generation method, we cannot rely on established baselines. Instead, we design the following custom baselines. First, we evaluate whether macrocycles can be generated by **finetuning** existing models on macrocyclic datasets (Appendix H.1). Second, we construct a **naive** guidance mechanism that forces a chosen subset of atoms to form a ring, where adjacent atoms are kept close and opposite atoms are pushed apart (Appendix H.2). We check the performance for subset sizes from 12 to 16 atoms. Finally, we test whether initializing the denoising process with **torus-shaped noise** resembling a macrocycle improves generation (Appendix H.3).

**Metrics.** For consistency with previous approaches (Peng et al., 2023), we reuse the following metric definitions: molecules are considered **valid** if they can be parsed by RDKit (Landrum, 2013), **connected** if they have only one graph connected component, and **successful** if they meet both properties at the same time. If the generated molecule is successful, we test if the molecule is a **macrocycle** by checking for the presence of a chordless cycle of size at least 12 using `Chem.GetSymmSSSR` from RDKit. This only includes cycles without edges between non-neighboring vertices, which excludes fused small rings where the outer boundary can have $\geq 12$ atoms.

**Parameter choice.** During the experiments, we set $\lambda_t = 1$, $[d_{\min}, d_{\max}] = [4.45 \text{ Å}, 5.05 \text{ Å}]$, and $\ell^\star = 2$ Å, which is chosen based on chemical knowledge and is slightly higher than longest common bonds in organic chemistry.

**Results.** Table 1 demonstrates that MACROGUIDE increases the number of generated macrocycles by almost 20-fold compared to MolDiff, and outperforms all the other baselines. Finetuning does yield a respectable generation rate, although generating about 15% fewer macrocycles than MACROGUIDE. Visualizations of generated molecules are provided in Appendix I.1.

### 4.1.2. PROTEIN CONDITIONING

MACROGUIDE can also be applied to molecular generation models conditioned on proteins which we demonstrate by guiding MolSnapper (Ziv et al., 2025). MolSnapper requires specifying the protein pocket placement and choosing a few atoms from a reference ligand to act as a pharmacophore. We follow the setup provided by the original work: we use the same protein pocket and choose reference atoms to guide generation within the protein. We reuse the same parameters as in Section 4.1.1. Further setup details are described in Appendix G.3.

**Baselines.** For comparison, we include the same baselines as in Section 4.1.1, albeit without the torus noise initialization because of its poor performance in the unconditional setting and because of the non-triviality of choosing the right placement and angle of the initial macrocycle-like noise. We use the same finetuned checkpoint as before, thanks to architectural similarity between MolDiff and MolSnapper.

**Results.** We report performance in Table 2. Similarly to the unconditional setting, the macrocycle generation rate increases from 0.3% to nearly 100%, representing a 300-fold increase. Importantly, MACROGUIDE is the only efficient method, as all of the baselines exhibit a dramatic performance drop. In particular, the performance of the finetuned model drops to 18%. This collapse can be attributed to the protein constraints pushing the finetuned model outside of its training distribution, leaving no explicit mechanism to constrain the trajectory back towards macrocyclic topologies. We analyze this distinction as well the learnt feature space in detail in Appendix K: MACROGUIDE converges to macrocyclic topology within 100 steps and resists perturbations, while finetuning encodes only a slow, fragile implicit bias that is easily overridden by protein-pocket constraints. Consequently, we do not consider finetuning to be a robust solution for this task and do not assess it further. Visualizations of generated molecules are provided in Appendix I.2.

### 4.1.3. ASSESSING MACROCYCLE QUALITY

To further assess the quality of successful macrocycles generated with MACROGUIDE we evaluate the following additional criteria: **novelty**, indicating whether a molecule is absent from the training dataset of the original model; **uniqueness**, measuring whether it differs from all the other generated samples; and **diversity**, assessed via fingerprint similarity across all pairs of generated molecules (Peng et al., 2023). Finally, to assess the chemical quality of the generated macrocycles we run **PoseBusters metrics** (Buttenschoen et al., 2024) and report the fraction of molecules that pass all tests in the suite. We also show the performance for individual checks, omitting the ones with perfect or near-perfect performance.

*Table 1.* **Performance of unconditional macrocycle generation.** Results obtained from 1000 molecules with 30 heavy atoms.

| Metrics (↑; [0-1]) | MolDiff (no guid.) | MolDiff (finetuned) | Naive (12 atoms) | Naive (14 atoms) | Naive (16 atoms) | Torus noise initialization | MolDiff+MACROGUIDE (ours) |
|---|---|---|---|---|---|---|---|
| Validity | 0.991 | **0.994** | 0.937 | 0.930 | 0.918 | 0.993 | 0.989 |
| Connectivity | 0.958 | 0.993 | 0.985 | 0.990 | 0.987 | 0.960 | **0.999** |
| Successfulness | 0.949 | 0.987 | 0.923 | 0.921 | 0.906 | 0.953 | **0.988** |
| Out of successful: | | | | | | | |
| **Macrocycles** | 0.053 | 0.851 | 0.370 | 0.364 | 0.369 | 0.054 | **0.997** |

*Table 2.* **Performance of macrocycle generation with protein conditioning.**

| Metrics (↑; [0-1]) | MolSnapper (no guid.) | MolSnapper (finetuned) | Naive (12 atoms) | Naive (14 atoms) | Naive (16 atoms) | MolSnapper+MACROGUIDE (ours) |
|---|---|---|---|---|---|---|
| Validity | 0.858 | 0.795 | 0.819 | 0.791 | 0.763 | **0.925** |
| Connectivity | 1.000 | 1.000 | 1.000 | 1.000 | 1.000 | 1.000 |
| Successfulness | 0.858 | 0.795 | 0.819 | 0.791 | 0.763 | **0.925** |
| Out of successful: | | | | | | |
| **Macrocycles** | 0.003 | 0.180 | 0.013 | 0.000 | 0.003 | **0.995** |

*Table 3.* **Unconditional macrocycle quality metrics.** Values were obtained by generating molecules until at least 1000 macrocycles were found to ensure reliable and comparable estimates.

| Metrics (↑; [0-1]) | MolDiff (no guid.) | +MACROGUIDE (ours) |
|---|---|---|
| Diversity | 0.707 | **0.771** |
| Novelty | 1.000 | 1.000 |
| Uniqueness | 1.000 | 1.000 |
| **All PoseBusters tests** | 0.663 | **0.805** |
| Bond lengths | 0.977 | **0.990** |
| Bond angles | 0.949 | **0.987** |
| Internal steric clash | 0.722 | **0.844** |
| Aromatic ring flatness | 0.980 | **0.999** |
| Non-ar. ring non-flatness | 0.981 | **0.999** |
| Double bond flatness | 0.965 | **0.989** |
| Internal energy | 0.961 | **0.984** |

*Table 4.* **Protein-conditioned macrocycle quality metrics.**

| Metrics (↑; [0-1]) | MolSnapper (no guid.) | +MACROGUIDE (ours) |
|---|---|---|
| Diversity | 0.626 | **0.712** |
| Novelty | 1.000 | 1.000 |
| Uniqueness | 1.000 | 1.000 |
| **All PoseBusters tests** | 0.440 | **0.575** |
| Ligand PoseBusters | 0.539 | **0.626** |
| Bond lengths | 0.844 | **0.860** |
| Bond angles | **0.913** | 0.888 |
| Internal steric clash | **0.862** | 0.854 |
| Aromatic ring flatness | **0.996** | 0.992 |
| Non-ar. ring non-flatness | 0.993 | **0.999** |
| Double bond flatness | 0.980 | **0.993** |
| Internal energy | 0.818 | **0.921** |
| Protein PoseBusters | 0.806 | **0.911** |
| Protein-ligand max. distance | 1.000 | 1.000 |
| Min. distance to protein | 0.806 | **0.907** |
| Volume overlap with protein | 1.000 | 1.000 |
| Pharmacophore satisfaction | 0.769 | **0.789** |
| Macrocycle Lipinski | 0.551 | **0.638** |

**Conditional metrics.** For conditional generation we also calculate PoseBusters metrics related to the protein. Additionally, **pharmacophore satisfaction** is evaluated by matching each reference atom to the closest atom in the generated molecule. A match is valid if the atom types agree and the distance is within 1 Å (Ziv et al., 2025). A pharmacophore is considered satisfied if at least 80% of the reference atoms are matched. Finally, we assess whether the generated molecules are likely to be orally available. Although there are no clearly established **Lipinski rules for macrocycles**, we follow Garcia Jimenez et al. (2023) and Viarengo-Baker et al. (2021) to propose the following rules: Molecular Weight (MW) < 1000 Da, number of Hydrogen Bond Donors (HBD) ≤ 7 and lipophilicity between 2.4 and 6, and we report the fraction of macrocycles meeting all of the criteria.

**Results.** Table 3 and Table 4 summarize the quality metrics of the generated macrocycles, with standard deviations reported in Appendix J.1. MACROGUIDE substantially outperforms the base model for most of the tested metrics, both in the unconditional and conditional settings. Additionally, our protein-conditioned macrocycles exhibit an important improvement in the internal energy test, which checks pose deviation from the relaxed molecule conformation. This suggests that MACROGUIDE addresses one of the prominent challenges in protein-conditioned molecular design: generated molecules are often not in their energetic minimum, resulting in low binding affinities and changes in pose after redocking (Harris et al., 2023). Finally, we observe an increase in the rate of pharmacophore satisfaction and compliance with Lipinski rules for macrocycles.

Furthermore, we examine atom type and ring size distributions of generated molecules, and show that MACROGUIDE preserves molecule complexity during macrocycle generation (Appendix J.7 and J.8). We also show that MACROGUIDE macrocycles tend to have fewer small strained rings, whose presence often poses a synthetic challenge.

**Discussion.** Notably, MACROGUIDE does not explicitly guide towards better 3D structure or molecular properties. The observed improvement in generative metrics possibly stems from the fact that MACROGUIDE fixes the global shape of the molecule at early stages of the denoising process, which allows the denoiser to spend more time refining the local structure of the molecule (see Figure 7). In contrast, without guidance the macrocycles represent a deviation from the training distribution, and likely assume cyclical shapes only later during the generation process.

### 4.2. Additional MACROGUIDE Applications

#### 4.2.1. BICYCLIC MOLECULE GENERATION

Macrocycles can achieve better selectivity than small molecules, but as the ring grows, so does the conformational space of macrocycles, potentially allowing for off-target interactions. One way to generate larger binders without reducing selectivity is to generate bicyclic compounds (Heinis et al., 2009; Rowland et al., 2025). To generate bicycles, we modify the topological guidance to optimize the two largest $H_1$ components, and we keep the same desired death time range. In Table 5 we show 97% and 90% bicyclic performance rates in the unconditional and conditional setups respectively. An example of a generated bicyclic molecule is shown in Figure 2.

*Table 5.* **Performance of bicyclic molecule generation in unconditional and conditional settings.** A molecule is considered bicyclic if it has at least two chordless cycles of at least 12 atoms each. Results obtained from 1000 molecules with 45 heavy atoms.

| Metrics (↑; [0-1]) | MolDiff +MACROGUIDE | MolSnapper +MACROGUIDE |
|---|---|---|
| Validity | 0.998 | 0.821 |
| Connectivity | 0.931 | 1.000 |
| Successfulness | 0.929 | 0.821 |
| Out of successful: | | |
| **Bicyclic molecules** | 0.970 | 0.895 |

#### 4.2.2. $H_0$ DEATH GUIDANCE IMPROVES LARGE MOLECULE GENERATION

While originally designed as a stability safeguard for our guidance method, ablation studies reveal that $H_0$ death optimization improves large molecule generation beyond the macrocycle task. Notably, existing methods struggle with the generation of large molecular structures (Le et al., 2024) and often produce several disconnected components.

We show that it is possible to preserve molecule connectivity as the number of atoms increases, by applying our $H_0$ death optimization (Equation (5) with the gradient masking detailed in Appendix D). The results in Figure 4 show a drastic improvement in connectivity (80% versus 20% for $n = 110$). $H_0$ optimization therefore appears as an attractive sampling-time guidance method when dealing with large molecules.

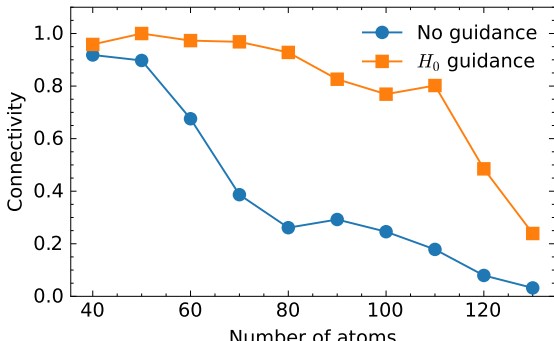

*Figure 4.* **Performance of MolDiff with increasing molecular size.** Adding $H_0$ guidance term improves performance for large molecule sizes. Results obtained from 200 samples each.

### 4.3. Further Analysis

#### 4.3.1. EMPIRICAL MACROCYCLE SIZE CONTROL

We empirically evaluate the results of Section 3.3 on macrocycle size control. Figure 5 confirms the validity of the provided control equation, although we observe a slight underestimation of the actual values by the theoretical predictions. One possible explanation is that elongated elliptic rings have a lower death time than circular rings with the same number of atoms (see Figure 9b of Appendix D for a visualization).

#### 4.3.2. RUNTIME COMPARISON

Importantly, MACROGUIDE introduces only a very moderate computational overhead on top of the base denoising process. Figure 6 provides a breakdown of the average time per denoising step for MolSnapper ($\approx 17ms$) and MACROGUIDE ($\approx 21ms$). The extra cost of our guidance method is dominated by the computation of the Vietoris-Rips complex, which grows quadratically with $n$. This is reasonable since the denoising process is quadratic too and MACROGUIDE represents only 20% of the total cost. Nevertheless, we explored two different strategies to accelerate our guidance method. Both methods rely on selecting only a subset of the denoising steps to apply the guidance function.

**Guidance every $k$ steps.** Our first strategy consists of applying the topological guidance only once every $k$ denoising steps, instead of at every step. Results for $k = 2, 3, 4, 5$ are reported in Appendix J.9.1. Importantly, we are able to

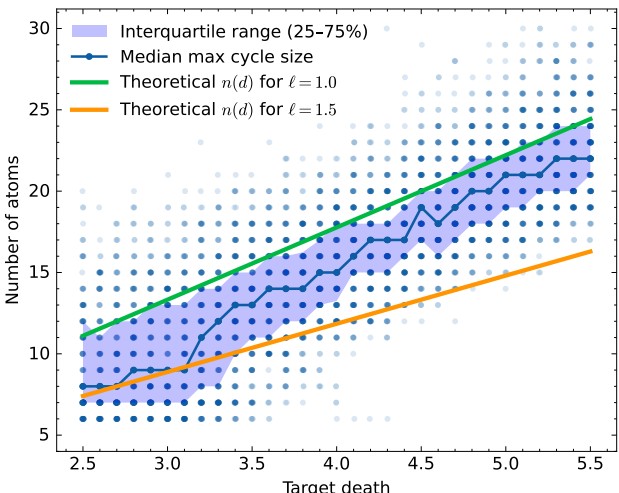

Figure 5. **Median max cycle size as a function of the target death.** The empirical results are compared to the theoretical formulas for $\ell = 1.0$ and $\ell = 1.5$, the minimum and maximum typical bond lengths, respectively. Results are computed for 200 samples of 30 heavy atoms, with each target size $d^\star$ being constrained in the relaxed form of an interval $[d^\star - 0.05, d^\star + 0.05]$, sampled at 0.1 intervals.

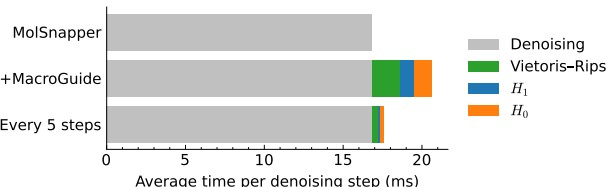

Figure 6. **Runtime comparison.** MACROGUIDE produces only a small computational overhead that can be further reduced by applying guidance only every $k$ steps.

divide the additional cost of MACROGUIDE by at least 5 with less than 15% degradation in macrocycle generation rates and molecular quality metrics relative to full guidance. Notably, it also improves the rate of pharmacophore satisfaction, compared both to MolSnapper and MACROGUIDE with $k = 1$. This trade-off makes sparse guidance an attractive default choice in practice.

**Late-start guidance.** A second strategy we investigated consists of activating the topological guidance only during the later stages of the diffusion process. In practice, however, this approach consistently underperformed the previous strategy, which can be explained by the fact that noising processes tend to corrupt high-frequency information first (Falck et al., 2025), which means the shape of a molecule is likely decided early during denoising. For completeness, we report the corresponding results in Appendix J.9.2.

Consequently, applying guidance every $k$ steps appears to be a good practical choice when trying to sample a handful of molecules, or when using high values of $n$.

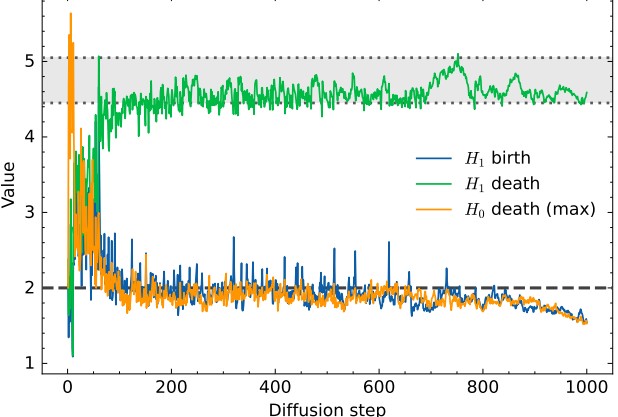

Figure 7. **Key topological features throughout denoising.** Maximum $H_0$ death (i.e. the longest distance to the closest neighbor over all atoms), and birth and death of the maximum-death $H_1$ component (largest cycle).

### 4.3.3. STABILITY ANALYSIS

The considerable sparsity of gradients may cast doubt on the stability of the method. That is why we provide insights about the guidance process by tracking optimized topological features and their gradients throughout denoising. Full results and discussion can be found in Appendix E. In particular Figure 7 demonstrates the stability of MACROGUIDE, after as few as 100 denoising steps. More importantly, it provides an explanation to the success of our method: guidance shapes macrocyclic structure early, while late-stage denoising (last 200 steps) performs fine-grained chemical validity optimization without interference.

### 4.4. Ablations and Other Experiments

We further analyze the contribution of the different components of our guidance mechanism through a series of ablations. In Appendix J.2, we remove individual terms of the guidance function and show that all components are necessary in the unconditional setting. In the conditional setting, however, the $H_0$ death term (molecule connectivity) can be omitted, as the confined volume of the protein pocket naturally prevents atom disconnection. In Appendix J.3, we investigate how adding or removing the square of the topological terms influences performance. We also study the sensitivity to guidance strength motivating the chosen value (Appendix J.4). We further motivate the use of Vietoris-Rips filtration compare to Alpha filtration (Appendix J.5). Additionally, we evaluate the robustness of MACROGUIDE to degraded base models by injecting Gaussian noise into pretrained MolDiff weights, demonstrating that it decouples global topology from local chemistry (Appendix J.6). Finally, we apply MACROGUIDE to flow matching formulations and observe a significant increase in macrocycle generation rate (Appendix J.10).

# 5. Conclusion

In this work, we presented MACROGUIDE, the first diffusion-based framework designed to generate arbitrary macrocycles without the need for model retraining or specialized datasets. By leveraging persistent homology to guide pretrained models, our method successfully bridges the gap between local chemical validity and global topological constraints. Specifically, our method drives the denoising path to a stable region of the set of cyclic structures, where validity optimization can be performed during later steps without interference from the guidance terms.

Experimentally, MACROGUIDE achieves a 99% macrocycle generation rate in both unconditional and protein-conditioned settings. Furthermore, the resulting molecules demonstrate superior 3D structural quality, exceeding or matching state-of-the-art performance on chemical validity, diversity, and PoseBusters benchmarks.

By providing a lightweight and training-free mechanism, MACROGUIDE offers a robust solution for exploring the macrocyclic chemical space. We provide a detailed discussion of promising future directions in Appendix B. We anticipate that this framework will integrate naturally with the expanding class of 3D point-cloud-based models, allowing topological guidance to contribute to continued improvements in generative drug discovery.

## Acknowledgements

The authors would like to thank Aleksy Kwiatkowski for insightful discussions. AM is partially funded by AITHYRA and the UKRI Engineering and Physical Sciences Research Council (EPSRC) with grant code EP/S024093/1. This research is partially supported by the EPSRC Turing AI World-Leading Research Fellowship No. EP/X040062/1 and EPSRC AI Hub No. EP/Y028872/1.

## Impact Statement

This work aims to improve controllable molecular generative modeling for drug discovery. By enabling more structured exploration of chemical space, such methods have the potential to accelerate early-stage drug design, reduce the cost of candidate discovery, and support the development of novel therapeutics, including for diseases with limited or no existing treatments and for traditionally challenging or undruggable targets. Improved generative efficiency may also help lower barriers to early-stage molecular design, enabling broader access to computational drug discovery tools across research communities.

At the same time, as with any general-purpose molecular generation approach, there are potential risks. Techniques that facilitate the design of therapeutic molecules could also be misused to generate harmful or toxic compounds. This work focuses on methodological development rather than deployment, and practical applications require appropriate domain expertise, experimental validation, and regulatory oversight. Generative models should therefore be viewed as decision-support tools rather than autonomous systems for drug design.

Overall, we believe the potential benefits of these methods for drug discovery outweigh the associated risks when applied responsibly within established scientific and ethical frameworks.

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

# A. Additional Related Literature

## A.1. Cyclic Peptide Generation

Most existing approaches focus on a restricted subclass of macrocycles, called cyclic peptides, where all building blocks are amino acids linked by peptide bonds. AfCycDesign adapts the AlphaFold framework (Jumper et al., 2021) by encoding cyclicity through a modified positional encoding, enabling the generation and structural prediction of cyclic peptides (Rettie et al., 2025a). Similarly, RFpeptides builds on RFdiffusion (Watson et al., 2023) and introduces a cyclic relative positional encoding to generate macrocyclic peptide backbones (Rettie et al., 2025b). Other works instead incorporate explicit geometric constraints or harmonic SDE formulations (Jiang et al., 2025; Zhou et al., 2025). However, the design space of cyclic peptides is substantially smaller and more structured than that of general macrocycles, making the generation task comparatively easier. As a result, these methods do not readily extend to the broader and more diverse class of non-peptidic macrocycles.

Additionally, a closely related problem is the prediction of 3D macrocyclic conformations from a given 2D molecular graph. This task was addressed by RINGER, which introduced a diffusion-based transformer model to generate ensembles of 3D conformations for macrocyclic peptides (Grambow et al., 2023). More recently, PuckerFlow proposed a flow-matching approach operating in Cremer-Pople space to generate conformers of small cyclic molecules (Schaufelberger et al., 2026).

## A.2. Applications of Persistent Homology in Machine Learning

**General use of persistent homology in ML.** Persistent homology (PH) has established itself as a critical framework in machine learning for characterizing data topology across multiple scales. Early integration strategies focused on creating differentiable, topology-aware architectures and losses. Notably, Topology Layer (Gabrielsson et al., 2020) and PersLay (Carrière et al., 2020) allow deep neural networks to directly process persistence diagram features. PHom-GeM (Charlier et al., 2019) instead introduced a topological loss function based on the bottleneck distance, ensuring that generated distributions faithfully reproduce the topology of the training set.

**Applications to diffusion models.** More recently, topological insights have been effectively adapted for diffusion-based generative frameworks across image, 3D, and graph domains. In the context of 3D and visual generation, Topology-Aware Latent Diffusion models (Hu et al., 2024) leverage PH to guide the formation of 3D shapes with specific topological characteristics. Similarly, Wang & Berger (2025) demonstrated that diffusion models can be steered using coordinate-based networks to satisfy rigorous topological constraints, while TopoDiffusionNet (Gupta et al., 2025) introduced a differentiable mechanism to regulate the number of objects (0-dim) and holes (1-dim) in generated image masks. In the graph domain, PH is increasingly used to capture complex connectivity; Park et al. (2025) developed a loss function based on persistence diagram matching for graph diffusion, and HOG-Diff (Huang & Birdal, 2025) introduced a framework for generating higher-order structures via cell complex filtering and spectral diffusion.

**Molecular applications.** In the realm of chemistry, persistent homology has proven effective for encoding molecular structure into ML-friendly representations. Townsend et al. (2020) introduced persistence images, 2D embeddings of PH-derived features, as molecular fingerprints to screen vast chemical libraries like GDB-9. Later, Demir & Kiziltan (2023) extended this with multiparameter PH fingerprints, incorporating atomic descriptors such as partial charges and bond types to enhance property prediction tasks. Finally, TDA has been used to augment a VAE with topological molecule representations which resulted in an improvement over a range of molecular generation metrics (Schiff et al., 2022).

# B. Discussion and Future Work

**Leveraging base model advancements.**   Since MACROGUIDE enhances generation metrics, it is well-positioned to benefit from rapid advancements in base model architectures. A promising avenue for future work is the integration of MACROGUIDE with recent, high-capacity models, potentially unlocking even greater performance.

**Bond-level information.**   Our method relies on atomic positions to enforce macrocyclicity, without explicitly constraining the neighboring atoms in the cycle to form bonds. This design naturally aligns with point-cloud-based models such as EDM, which do not represent bonds explicitly (Hoogeboom et al., 2022). For architectures that additionally model bond information (e.g., MolDiff), one could in principle incorporate bond-level constraints into the guidance. Our preliminary investigations (Appendix L) suggest that the added complexity of this approach outweighs the potential benefits, though it remains an open option for future work.

**Explicit hydrogens.**   We apply MACROGUIDE to models that do not explicitly model hydrogens during training, a common setting in recent works, as the hydrogens can be easily inferred during post-processing to avoid unnecessary time and memory overhead. However, if one chooses to work with a model that represents hydrogens explicitly, their monovalence can hinder cycle closure. In this case, a simple workaround is to exclude hydrogens from the Vietoris-Rips complex.

**Synthetic accessibility.**   Macrocycles offer attractive pharmacological properties, but their synthesis typically involves an additional entropic cost associated with ring closure. As macrocycle-generating models move closer to practical deployment, incorporating synthetic considerations will become increasingly important. A natural and promising next step is to integrate synthesis-aware constraints and retrosynthetic planning into the generative process, enabling closer alignment between generated candidates and experimentally accessible chemistries.

## C. Proof of Theorem 3.1

**Lemma C.1** (Vietoris-Rips death time of a regular n-gon). *Consider a regular n-gon of side length $\ell$. The death time d of the dominant $H_1$ component in the Vietoris-Rips filtration is given by:*

$$d = \ell \frac{\sin\left(\left\lceil \frac{n}{3} \right\rceil \frac{\pi}{n}\right)}{\sin\left(\frac{\pi}{n}\right)}$$

*Proof of Lemma C.1.* Following the theory of Vietoris-Rips complexes of the circle established by Adamaszek & Adams (2017), the homotopy type of $VR(X;r)$ for a finite subset $X \subset S^1$ is determined by its winding fraction $wf(X;r)$. For a regular $n$-gon with side length $\ell$, the complex is equivalent to the clique complex of the cyclic graph $C_n^k$.

**Threshold for death:** The $H_1$ component persists as long as the winding fraction satisfies $0 < wf < 1/3$.

**Geometric jump:** For a regular $n$-gon, the winding fraction is defined as $wf = k/n$, where $k$ is the maximum number of edges a single chord of length $d$ can span. Thus, death occurs at the smallest $d$ such that $k = \lceil n/3 \rceil$.

**Exact formula:** Let $R$ be the circumradius of the $n$-gon. The side length $\ell$ and the death distance $d$ are chords related by:

$$\ell = 2R\sin\left(\frac{\pi}{n}\right), \quad d = 2R\sin\left(\lceil n/3 \rceil \frac{\pi}{n}\right)$$

Dividing these yields the exact relation:

$$d = \ell \frac{\sin\left(\left\lceil \frac{n}{3} \right\rceil \frac{\pi}{n}\right)}{\sin\left(\frac{\pi}{n}\right)}$$

$\square$

*Proof of Theorem 3.1.* **Extension to tetrahedral conformation:** Consider $n$ atoms in a zigzag cycle. Because the bond angles are constrained, the atoms alternate between two concentric radii. Death edges can only be between two of the $n/2$ atoms of the interior circle. These atoms form a planar regular $N$-gon where $N = n/2$. That means we can apply Lemma C.1. The proof geometry is illustrated in Figure 8.

**Effective side length:** The distance between two adjacent atoms in this sub-polygon is determined by the bond length $\ell$ and the bond angle $\theta$. More precisely the law of cosines gives

$$\ell' = \sqrt{\ell^2 + \ell^2 - 2\ell^2 \cos\theta} = \ell\sqrt{2(1 - \cos\theta)}$$

**Exact formula:** Substituting $N = n/2$ and $\ell'$ into the Lemma:

$$d = \ell\sqrt{2(1 - \cos\theta)} \cdot \frac{\sin\left(\left\lceil \frac{n}{6} \right\rceil \frac{2\pi}{n}\right)}{\sin\left(\frac{2\pi}{n}\right)}$$

**Linearization:** As $n \to +\infty$, we get

$$d \approx \ell\sqrt{2(1 - \cos\theta)} \frac{\sqrt{3}/2}{2\pi/n} = \frac{n\ell\sqrt{6(1 - \cos\theta)}}{4\pi}$$

Solving for $n$ yields the linear form:

$$n = \frac{4\pi}{\sqrt{6(1 - \cos\theta)}} \frac{d}{\ell} + \mathcal{O}(1)$$

$\square$

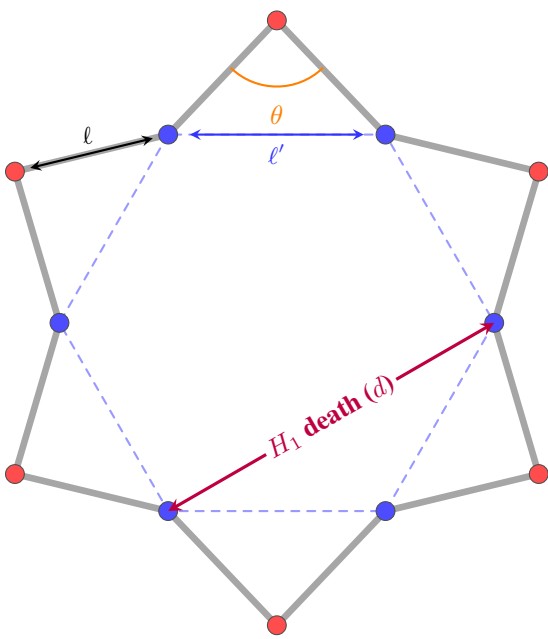

*Figure 8.* **Proof geometry: crown-shaped $n$-gon ($n = 12$).** The inner polygon (dashed blue) determines the filtration death time $d$. The effective side length $\ell'$ is derived from the bond length $\ell$ and the interior bond angle.

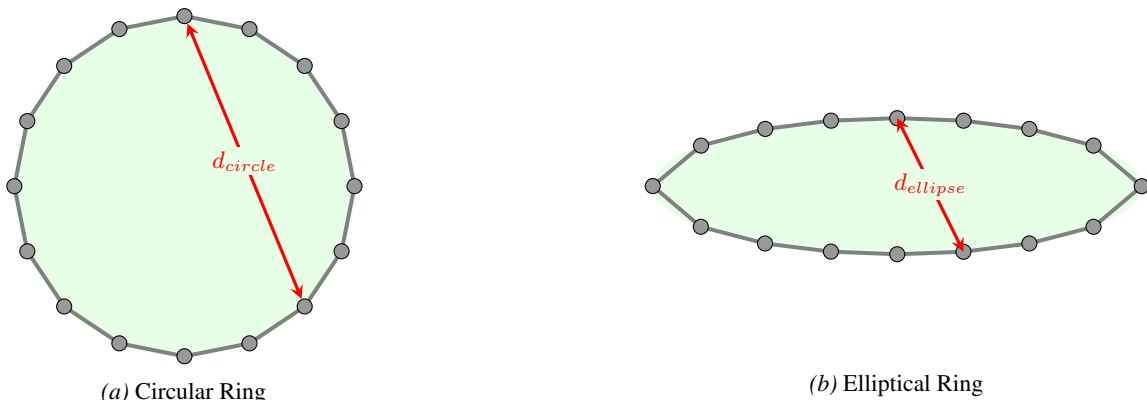

*(a)* Circular Ring

*(b)* Elliptical Ring

*Figure 9.* **Impact of cycle shape on $H_1$ death ($n = 16$).** In an elongated elliptical macrocycle, the death time $d_{ellipse}$ is significantly reduced compared to $d_{circle}$. The death times were obtained numerically using torch_topological.

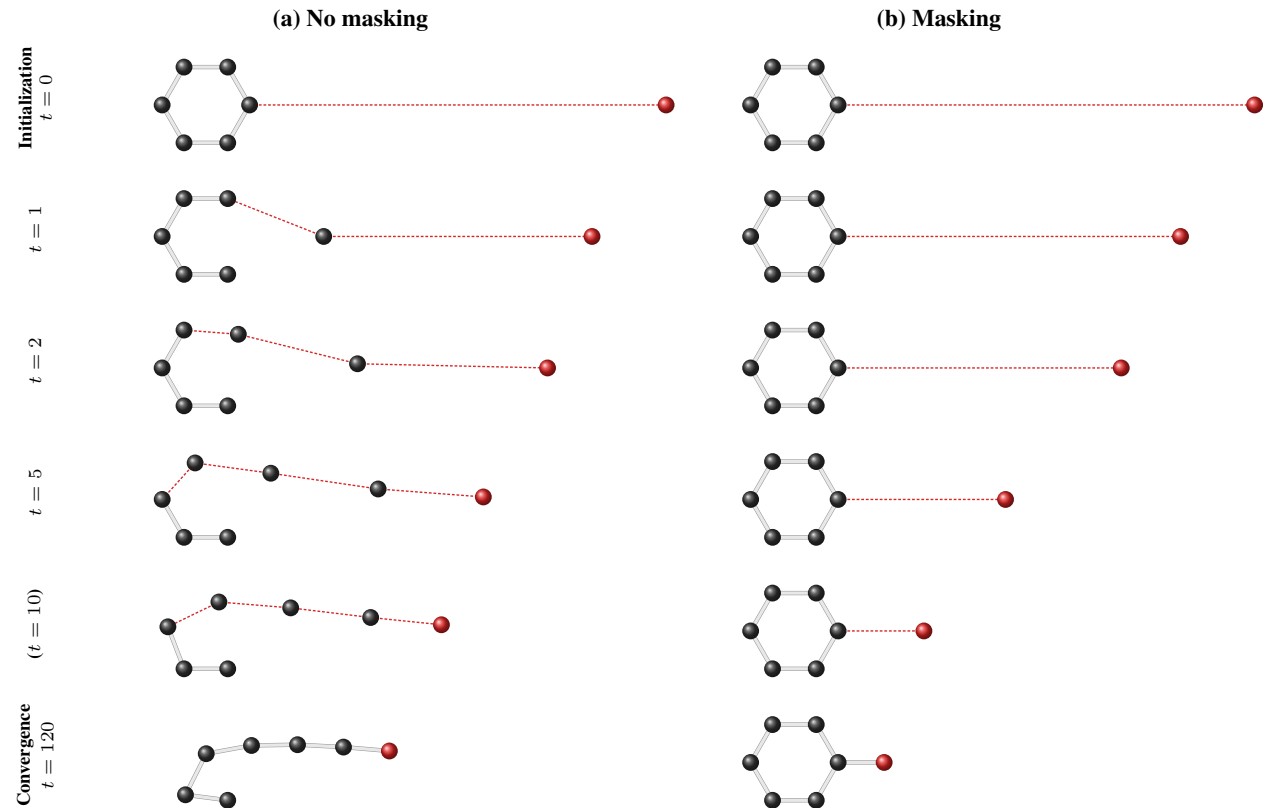

*Figure 10.* **Failure mode analysis.** Comparison of optimization dynamics under Equation (9). (a) Symmetric gradients cause "unzipping." (b) Masked gradients preserve structure.

# D. Details on $F_{\text{death}}^{H_0}$: Gradient Masking for Stability

**Method.** As presented in Section 3.1, connectivity is enforced by minimizing the $H_0$ persistence death times that exceed a threshold $\ell^\star$. In the Vietoris-Rips filtration, a finite $H_0$ death time corresponds to the scale at which two connected components merge. Formally, the connectivity loss is defined over the set of finite death times $\mathcal{D}_0(X)$ in the 0-dimensional persistence diagram:

$$F_{\text{death}}^{H_0}(X) = \sum_{d \in \mathcal{D}_0(X)} \left( \text{ReLU}\big(d - \ell^\star\big) \right)^2 \tag{9}$$

To optimize this objective, we introduce a **masked gradient** scheme. Each death value $d > \ell^\star$ is determined by the distance between two specific atoms $(u, v)$ that trigger the merging of two components (i.e., $\|x_u - x_v\| = d$). We define the centroid of the system as $\mathbf{c} = \frac{1}{N} \sum_{i=1}^{N} x_i$. The gradient applied to each endpoint of this critical edge goes through by a binary mask $m$:

$$g_u = m_u \cdot \nabla_{x_u} F_{\text{death}}^{H_0}, \quad m_u = \mathbb{1}\left( \|x_u - \mathbf{c}\| \geq \|x_v - \mathbf{c}\| \right)$$

In other words, for every pair of atoms responsible for the merging of two connected components (that are separated by more than $\ell^\star$), we block the gradient flow for the atom closer to the centroid ($m = 0$) and allow it only for the atom further away ($m = 1$).

**Instability of symmetric optimization.** The necessity of this masking becomes apparent when analyzing the case of a stable connected cluster $\mathcal{C}$ (stable in the sense that all the $H_0$ death times are less than $\ell^\star$) interacting with a distant outlier component $o$. Let's assume that the merging of the outlier with the cluster determines a death time $d = \|x_u - x_o\| > \ell^\star$, for some $u \in \mathcal{C}$ (i.e. the outlier is at a distance $d$ from the cluster). Without masking, the gradient of Equation (9) is applied symmetrically. The internal atom $u$ moves as

$$\Delta x_u \propto -\nabla_{x_u} F_{\text{death}}^{H_0} \propto (x_o - x_u)$$

Since $o$ is an outlier, this vector points away from the cluster centroid. Consequently, $u$ is pulled out of its local equilibrium within $\mathcal{C}$. This displacement stretches the distances between $u$ and its neighbors. If the force exerted by the outlier is sufficiently strong (i.e., if it is sufficiently far), these internal distances cross the threshold $\ell^\star$, creating new high-persistence $H_0$ features that trigger their own penalty terms in Eq. 9. This creates a cascading failure where the cluster is sequentially pulled apart, resulting in the *linearization* or "unzipping" of the cycle structure, as demonstrated in Figure 10a. By clamping the inner atom $u$ (via $m_u = 0$), our method decouples the outlier's transport from the cluster's internal dynamics, ensuring structural preservation (Figure 10b).

**Non-conservative field.** Note that this masking renders the resulting guidance vector field non-conservative, effectively acting as a stabilized approximation of the exact score function described in Equation (6). Nevertheless, the theoretical intuition provided in Section 3.2 remains relevant, as the masked update still locally maximizes the conditional likelihood $p_t(y|x)$ to drive the system toward the desired topology, differing only by the suppression of specific unstable gradient directions.

Importantly, non-conservative guidance does not preclude convergence. While a conservative drift guarantees convergence to the Boltzmann distribution, a non-conservative guidance field can still converge to a stationary distribution, provided the conservative score component of the base model prevents divergence. In our setting, convergence of the guided process to the target marginal distribution is sufficient for the desired generative performance, as evidenced by the rapid convergence of all topological features within the first 100 denoising steps (Figure 7, Appendix E).

## E. Topological Features and Their Gradients Throughout Denoising

To better understand the success of MACROGUIDE, we analyze the evolution of the topological features and their associated gradients throughout the 1000 steps of the reverse diffusion process.

**Feature convergence and constraint satisfaction.** As shown in Figure 11, the guidance successfully steers the molecule toward the target topology. All optimized features converge rapidly to their respective targets: the largest cycle size ($H_1$ death) reaches the target interval within the first few hundred steps, while cycle connectivity ($H_1$ birth) and molecular connectivity ($H_0$ death) are effectively kept below the threshold $\ell^\star$. Once satisfied, these constraints remain stable throughout the remainder of the denoising process, ensuring that the global macrocyclic structure is established early and preserved.

**Gradient analysis.** Figure 14 illustrates that, on average, the topological gradients are of the same magnitude as the denoising gradients. We observe that $H_1$ death gradients are dominant at the very beginning of the process, driving the initial ring expansion. The $H_1$ birth gradients, while sparse throughout the trajectory, are constant in norm by design, providing sharp corrections when edges exceed the threshold $\ell^\star$. Crucially, in the last 100 steps (where fine-grained chemical details such as bond orders and angles are finalized) the gradients are dominated almost exclusively by the denoising term. This explains the success of our method: the guidance effectively pushes the molecule into a stable macrocyclic state early on, allowing the denoiser to perform optimal local validity optimization in the final stages without interference.

**Comparison with squared $H_1$ birth.** We compare this behavior with the one achieved by squaring the formula of $F_{\text{birth}}^{H_1}$ as is done for the other guidance terms. As seen in Figure 13, the $H_1$ birth feature in the squared setting stays very close to the target $\ell^\star$ but tends to oscillate around it rather than remaining strictly below. This behavior is confirmed by the gradient analysis in Figure 15, which shows that $H_1$ birth gradients remain very significant and exceeds the denoising gradients, even during the last 100 steps. Following the reasoning above, this persistent perturbation interferes with the denoiser's validity optimization in the critical final phase, explaining the lower performance observed with the squared $H_1$ birth term.

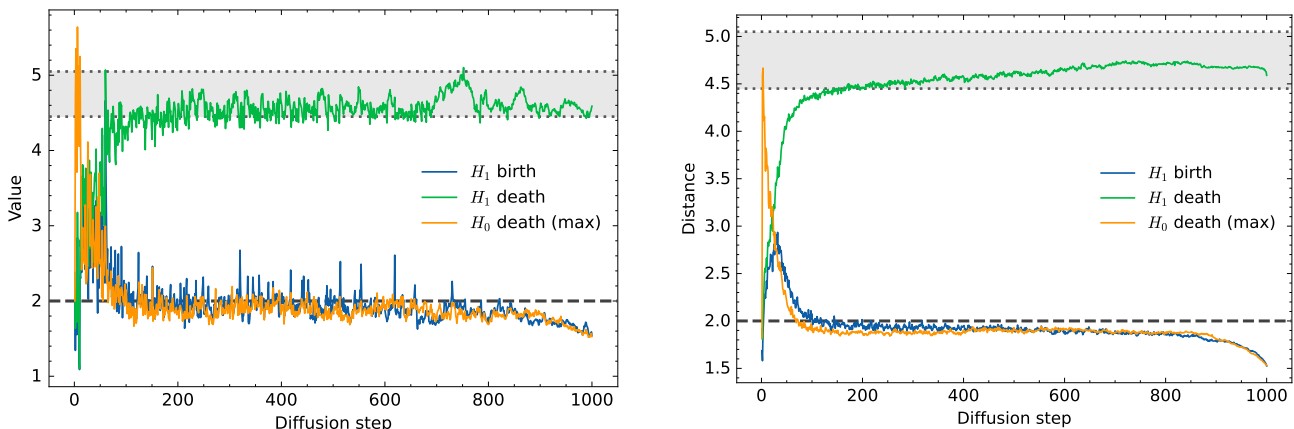

*Figure 11.* Topological features (no square on $F_{\text{birth}}^{H_1}$) for individual runs (left) and averaged runs (right).

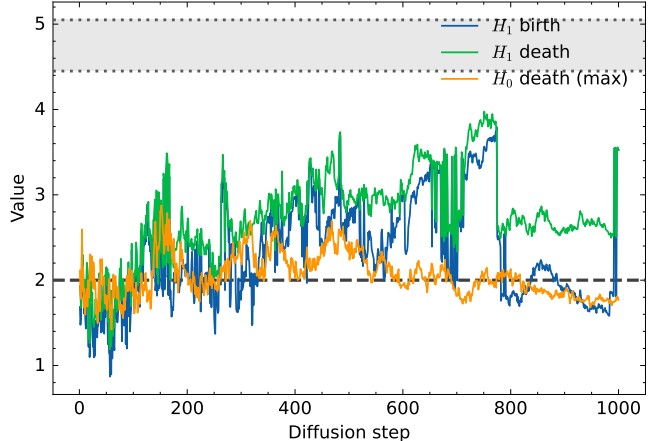

*Figure 12.* Topological features without guidance

## F. Detailed Algorithms

We present the detailed algorithms used by MACROGUIDE. Algorithm 1 shows how MACROGUIDE fits into the overall scheme of denoising. Algorithm 2 describes the high-level structure of MACROGUIDE, whereas algorithms 3 and 4 describe its components: $H_1$ and $H_0$ guidances, respectively.

**Relationship to continuous score matching.** We note a specific detail regarding the implementation of topological guidance in Algorithm 1 compared to the theoretical formulation in Equation (1). While Equation (1) defines guidance as a modification to the instantaneous score function $\nabla_{x_t} \log p_t(y|x_t)$, our algorithm implements this as a sequential predictor-corrector step. Specifically, we first apply the base denoising update to obtain an intermediate state $x_{t-1}$, and then calculate the topological gradient $\Delta$ on this cleaner state. In this discrete setting, the scalar weight $\lambda$ acts as an effective step size that implicitly absorbs the SDE integration constants (e.g., $\sigma_t^2$) required to map the score-space gradient to a coordinate-space displacement.

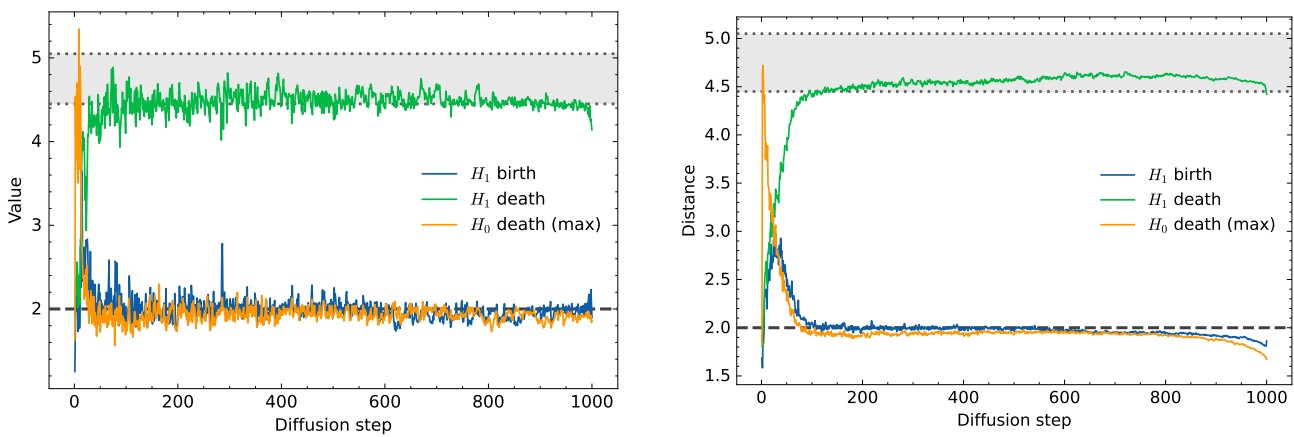

*Figure 13.* Topological features **with squared** $F_{\text{birth}}^{H_1}$ for individual runs (left) and averaged runs (right).

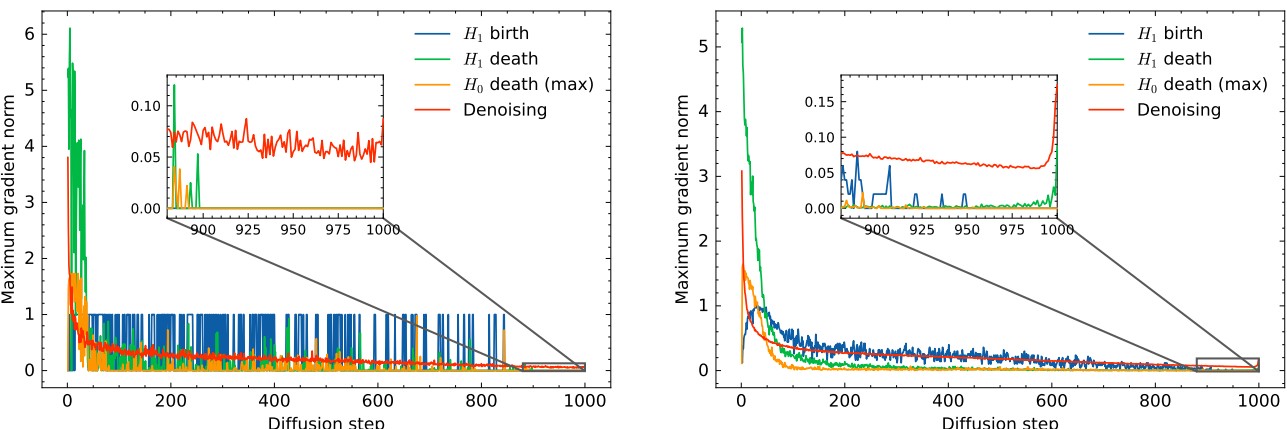

*Figure 14.* Gradient statistics (no square on $F_{\text{birth}}^{H_1}$) for individual runs (left) and averaged runs (right).

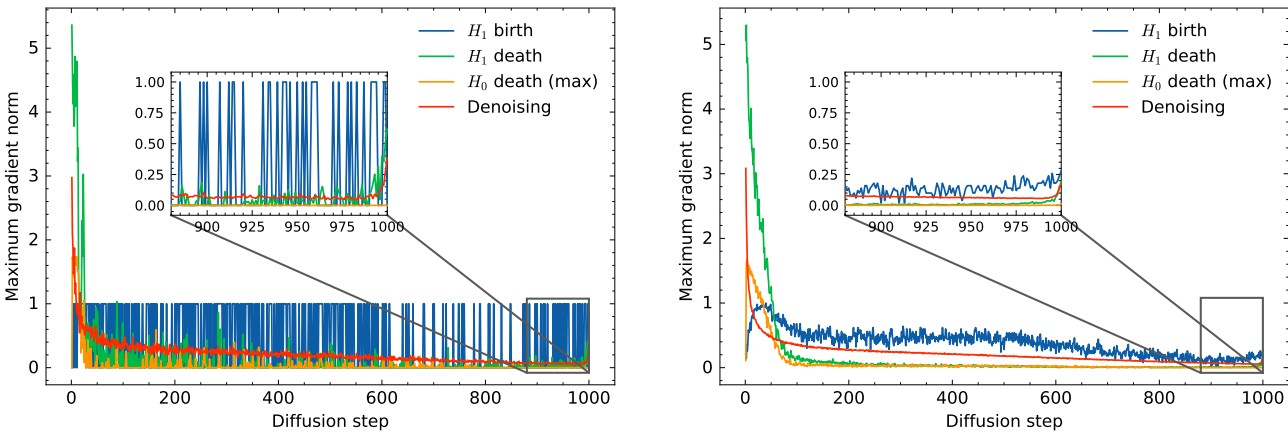

*Figure 15.* Gradient statistics **with squared** $F_{\text{birth}}^{H_1}$ for individual runs (left) and averaged runs (right).

---

**Algorithm 1** DENOISEWITHMACROGUIDE

---

**Input:** denoiser $\epsilon_\theta$; noise schedule $\{\sigma_t\}_{t=1}^{T}$; initial sample $\mathbf{x}_T$; guidance parameters $(\lambda, \ell^*, d_{\min}, d_{\max})$, guidance schedule $\mathcal{T}_g \subset \{1, \ldots, T\}$
**for** $t = T$ **down to** 1 **do**
    $\hat{\epsilon} \leftarrow \epsilon_\theta(\mathbf{x}_t, \sigma_t)$
    $\mathbf{x}_{t-1} \leftarrow$ DENOISEUPDATE$(\mathbf{x}_t, \hat{\epsilon}, \sigma_t)$
    **if** $t \in \mathcal{T}_g$ **then**
        $\Delta \leftarrow$ MACROGUIDE$(\mathbf{x}_{t-1}; \ell^*, d_{\min}, d_{\max})$ {Algorithm 2}
        $\mathbf{x}_{t-1} \leftarrow \mathbf{x}_{t-1} + \lambda \cdot \Delta$
    **end if**
**end for**
**return** $\mathbf{x}_0$

---

**Algorithm 2** MACROGUIDE

---

**Input:** Coordinates $\mathbf{X}$, $\ell^*, d_{\min}, d_{\max}$
$\Pi \leftarrow$ VietorisRips$(\mathbf{X})$
$\mathbf{g}_{H_1} \leftarrow H_1$GUIDANCE$(\mathbf{X}, \Pi, \ell^*, d_{\min}, d_{\max})$ {Algorithm 3}
$\mathbf{g}_{H_0} \leftarrow H_0$GUIDANCE$(\mathbf{X}, \Pi, \ell^*)$ {Algorithm 4}
$\Delta \leftarrow -(\mathbf{g}_{H_1} + \mathbf{g}_{H_0})$
**Return** $\Delta$

---

**Algorithm 3** $H_1$GUIDANCE

---

**Input:** coordinates $X$, persistence info $\Pi$, thresholds $\ell^\star, d_{\min}, d_{\max}$
Initialize objectives $F_{\text{birth}}^{H_1} \leftarrow 0, \quad F_{\text{death}}^{H_1} \leftarrow 0$
**for** each sample $k$ in batch **do**
    Extract $H_1$ diagram pairs $P$ from $\Pi_k$
    **if** $P$ is empty **then**
        **continue**
    **end if**
    {Identify the $H_1$ component (ring) that dies last}
    Find index $i^\star = \arg\max_i \text{death}(P_i)$
    Retrieve critical vertices for feature $i^\star$:
        $(u_b, v_b)$ defining birth time $b_{i^\star}^{(1)}$
        $(u_d, v_d)$ defining death time $d_{i^\star}^{(1)}$
    **1. Cycle Connectivity ($H_1$ birth):**
    $b_{i^\star}^{(1)} \leftarrow \|x_{u_b} - x_{v_b}\|$
    $F_{\text{birth}}^{H_1} \leftarrow F_{\text{birth}}^{H_1} + \text{ReLU}(b_{i^\star}^{(1)} - \ell^\star)$
    **2. Cycle Size ($H_1$ death):**
    $d_{i^\star}^{(1)} \leftarrow \|x_{u_d} - x_{v_d}\|$
    {Penalize if outside target interval $[d_{\min}, d_{\max}]$}
    **if** $d_{i^\star}^{(1)} < d_{\min}$ **then**
        $F_{\text{death}}^{H_1} \leftarrow F_{\text{death}}^{H_1} + (d_{\min} - d_{i^\star}^{(1)})^2$
    **else if** $d_{i^\star}^{(1)} > d_{\max}$ **then**
        $F_{\text{death}}^{H_1} \leftarrow F_{\text{death}}^{H_1} + (d_{i^\star}^{(1)} - d_{\max})^2$
    **end if**
**end for**
$\mathcal{F}_{\text{TDA}}^{(1)} \leftarrow F_{\text{birth}}^{H_1} + F_{\text{death}}^{H_1}$
Compute gradients $\mathbf{g}_{H_1} \leftarrow \nabla_X \mathcal{F}_{\text{TDA}}^{(1)}$
**Return** $\mathbf{g}_{H_1}$

---

---

**Algorithm 4** $H_0$GUIDANCE

---

**Input:** coordinates $X$, persistence info $\Pi$, threshold $\ell^\star$

Initialize connectivity objective $F_{\text{death}}^{H_0} \leftarrow 0$

**for** each sample pair $(X^{(k)}, \pi)$ in $(X, \Pi)$ **do**

    Extract finite death times $\mathcal{D}_0(X^{(k)})$ and corresponding pairs $P$ from $\pi$

    Identify active edges $E \leftarrow \{(u, v) \in P \mid \text{death}(u, v) > \ell^\star\}$

    **if** $E$ is empty **then**

        **continue**

    **end if**

    $\mathbf{c} \leftarrow \frac{1}{N} \sum_j x_j$ {Calculate centroid of molecule $k$}

    **for** each critical edge $(u, v)$ in $E$ **do**

        {Apply masked gradient: move only the atom furthest from centroid}

        **if** $\|x_u - \mathbf{c}\| > \|x_v - \mathbf{c}\|$ **then**

            $u$ is active ($m_u = 1$),    $v$ is masked ($m_v = 0$)

            $x_{active} \leftarrow x_u$,    $x_{masked} \leftarrow \text{stop\_gradient}(x_v)$

        **else**

            $v$ is active ($m_v = 1$),    $u$ is masked ($m_u = 0$)

            $x_{active} \leftarrow x_v$,    $x_{masked} \leftarrow \text{stop\_gradient}(x_u)$

        **end if**

        $d_j^{(0)} \leftarrow \|x_{active} - x_{masked}\|$

        $F_{\text{death}}^{H_0} \leftarrow F_{\text{death}}^{H_0} + \left(\text{ReLU}(d_j^{(0)} - \ell^\star)\right)^2$

    **end for**

**end for**

$\mathbf{g}_{H_0} \leftarrow \nabla_X F_{\text{death}}^{H_0}$

**Return** $\mathbf{g}_{H_0}$

---

# G. Experimental Details

## G.1. MolDiff Training Dataset

The training split used for MolDiff contains 231,521 data points from GEOM-Drugs, out of which only 323 molecules had a ring of at least 12 atoms (Axelrod & Gomez-Bombarelli, 2022; Peng et al., 2023).

## G.2. PoseBusters

The analysis was ran with PoseBusters version 0.4.6.

## G.3. Protein Conditioning with MolSnapper

Following the original setup, we selected a subset of atoms from the reference ligand to define a pharmacophore and used the provided protein pocket corresponding to PDB ID 1H00 (Ziv et al., 2025; Beattie et al., 2003). We included all nitrogen atoms from the reference ligand, as these are the most likely to participate in specific interactions with the protein pocket and thus provide a strong conditioning signal. In total, the pharmacophore consists of five atoms: three nitrogens and two carbons.

## G.4. Computational Resources

All experiments were conducted on NVIDIA A10 GPUs, except for finetuning which was performed on NVIDIA H100 GPUs.

## G.5.

# H. Additional Baselines

This section provides descriptions of the baselines used to benchmark MACROGUIDE in Tables 1 and 2.

## H.1. Finetuning

We finetune MolDiff using the publicly released pretrained checkpoint and continue training on a macrocycle-specific dataset (Peng et al., 2023). The finetuning procedure follows the original training protocol, and was run for 50,000 iterations. Due to the increased size of macrocyclic molecules relative to those in the original training set, which leads to higher memory consumption during training, we reduce the batch size from 256 to 64 to fit within hardware constraints.

The macrocyclic dataset is derived from Macrocycle-DB (Jiang et al., 2026), which contains 45,525 macrocyclic compounds with resolved three-dimensional structures. We follow the data processing pipeline of MolDiff (Peng et al., 2023) and apply the same filtering criteria, in particular removing molecules containing atom types not present in the pretrained model. After filtering, 40,496 molecules remain. We split the resulting dataset into training and validation sets using an 85:15 ratio, yielding 34,422 training molecules and 6,074 validation molecules, which are used to monitor and control the finetuning process.

Because MolSnapper uses similar underlying architecture and checkpoints as MolDiff, it is possible for us to use the same finetuned denoising network in both models.

## H.2. Naive Topological Guidance

**Naive geometric cycle guidance.** We introduce a simple geometric guidance term that biases generated molecular point clouds toward forming a single macrocyclic structure. Importantly, we do *not* assume that the atoms are ordered along a ring. Instead, given a molecule with at least $n$ atoms, we select an arbitrary subset of $n$ atoms (e.g., the first $n$ atoms in the representation) and impose a virtual cyclic ordering solely for the purpose of defining the guidance objective. This ordering has no chemical meaning and is used only to provide a differentiable proxy for cyclic topology.

Let

$$X = (x_0, \dots, x_{n-1}) \in \mathbb{R}^{n \times 3}$$

denote the coordinates of the selected atoms. We define two complementary geometric constraints: a local adjacency constraint encouraging ring closure, and a global separation constraint preventing degenerate, collapsed configurations.

**Local cyclic adjacency.** We enforce approximate uniformity of distances between consecutive atoms in the imposed cyclic order. Defining

$$\ell_i = \|x_i - x_{i+1}\|_2, \qquad x_n \equiv x_0,$$

we penalize distances that fall outside a target interval $[\ell_{\min}, \ell_{\max}]$. This is implemented using a hinge-squared penalty:

$$\mathcal{L}_{\text{edge}}(X) = \sum_{i=0}^{n-1} \Big[ \max(0, \ell_{\min} - \ell_i)^2 + \max(0, \ell_i - \ell_{\max})^2 \Big].$$

In our experiments, we set $\ell_{\min} = 1.0$ Å and $\ell_{\max} = 2.0$ Å. This term encourages the selected atoms to form a closed loop with approximately consistent edge lengths, without requiring exact bond formation or atom ordering.

**Global anti-collapse constraint.** To avoid degenerate solutions in which the loop collapses or folds onto itself, we introduce a coarse global constraint based on opposite atoms in the imposed cycle. For even $n$, we define

$$o_i = \|x_i - x_{i+n/2}\|_2,$$

and penalize opposite pairs that are closer than a minimum separation $o_{\min}$:

$$\mathcal{L}_{\text{opp}}(X) = \sum_{i=0}^{n-1} \max(0, o_{\min} - o_i)^2.$$

We set $o_{\min} = 3.0$ Å in all experiments. This term encourages a non-degenerate ring geometry with a finite diameter, discouraging self-intersections and collapsed configurations.

**Overall guidance objective.** The guidance is then given by the sum of $\mathcal{L}_{\text{edge}}$ and $\mathcal{L}_{\text{opp}}$. While this objective does not explicitly compute topological invariants such as persistent homology, it provides a computationally inexpensive and fully differentiable surrogate that promotes cycle-like geometries. In particular, it biases generation toward configurations exhibiting both local loop closure and global ring structure, despite the absence of any predefined atom ordering or explicit topological constraints.

### H.3. Solid-torus Noise Initialization

We replace standard Gaussian initialization with a prior distribution supported on a solid torus, defined by major radius $R = 2.5\,\text{Å}$ and tube radius $r_{\text{max}} = 1.5\,\text{Å}$. The coordinates $\mathbf{x} \in \mathbb{R}^3$ are sampled according to:

$$x = (R + \rho \cos \phi) \cos \theta,$$
$$y = (R + \rho \cos \phi) \sin \theta,$$
$$z = \rho \sin \phi,$$

where $\theta, \phi \sim \mathcal{U}[0, 2\pi)$ are uniform angular coordinates. The radial offset $\rho$ determines the deviation from the torus core and is sampled as $\rho = r_{\text{max}} \cdot \xi$, where $\xi \sim \text{Beta}(0.5, 2.0)$. This choice of hyperparameters ($\alpha < 1$) concentrates the probability mass near the central filament ($\rho \to 0$) while maintaining support throughout the tube.

**Prior mismatch.** Note that this initialization strategy fundamentally violates the stationary assumptions of the standard diffusion reverse process (Song et al., 2021), which is trained to map from an isotropic Gaussian prior. The negative results of Tables 1 and 2 are therefore not surprising.

# I. Additional Visualizations

## I.1. Unconditional Generation

Figure 16 displays molecules generated by MolDiff+MACROGUIDE, for the parameter values described in Section 4.1.1.

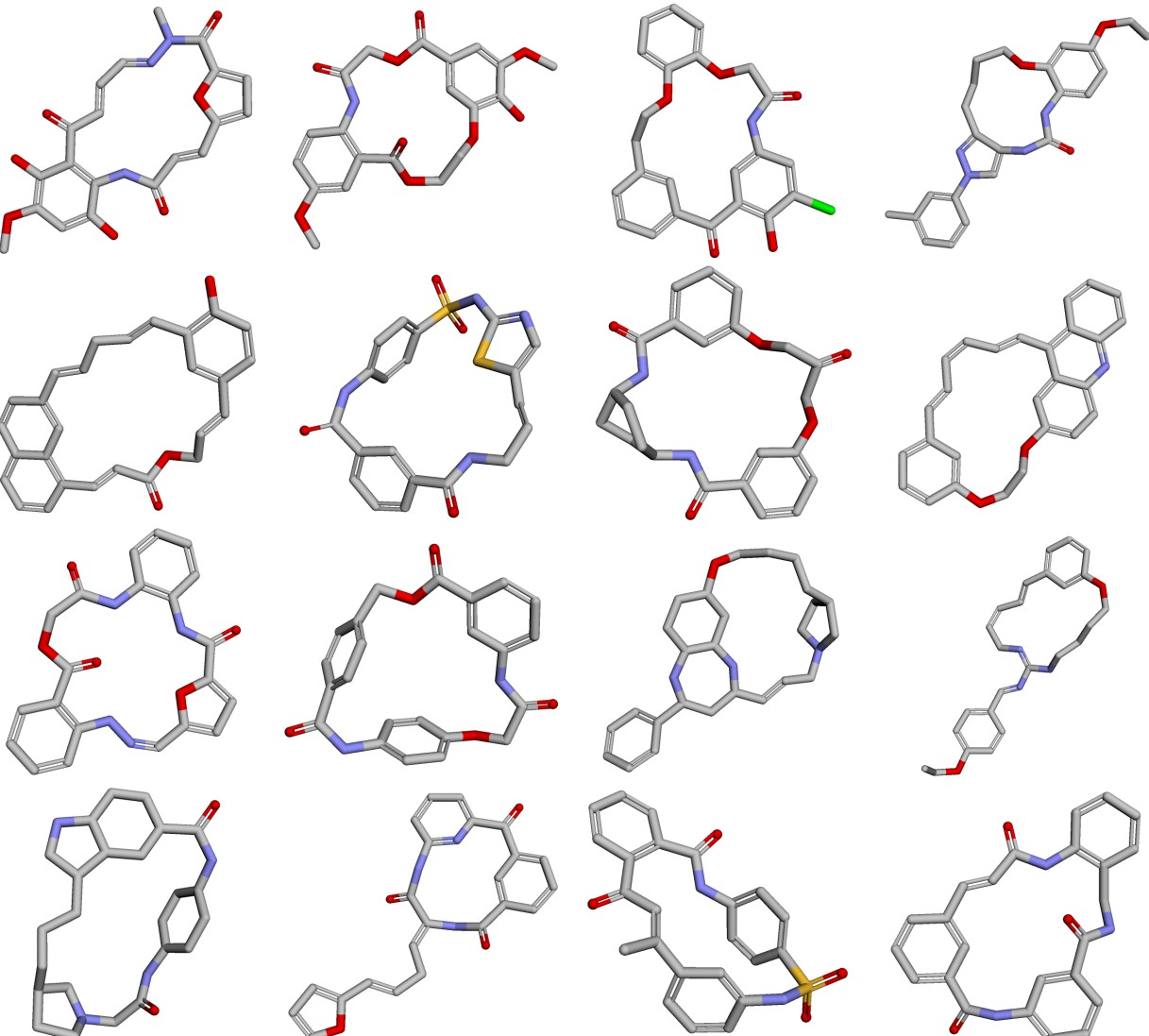

*Figure 16.* **Visualizations of generated molecules in the unconditional setting.**

## I.2. Conditional Generation

Figure 17 displays molecules generated by MolSnapper+MACROGUIDE, for the parameter values described in Section 4.1.2.

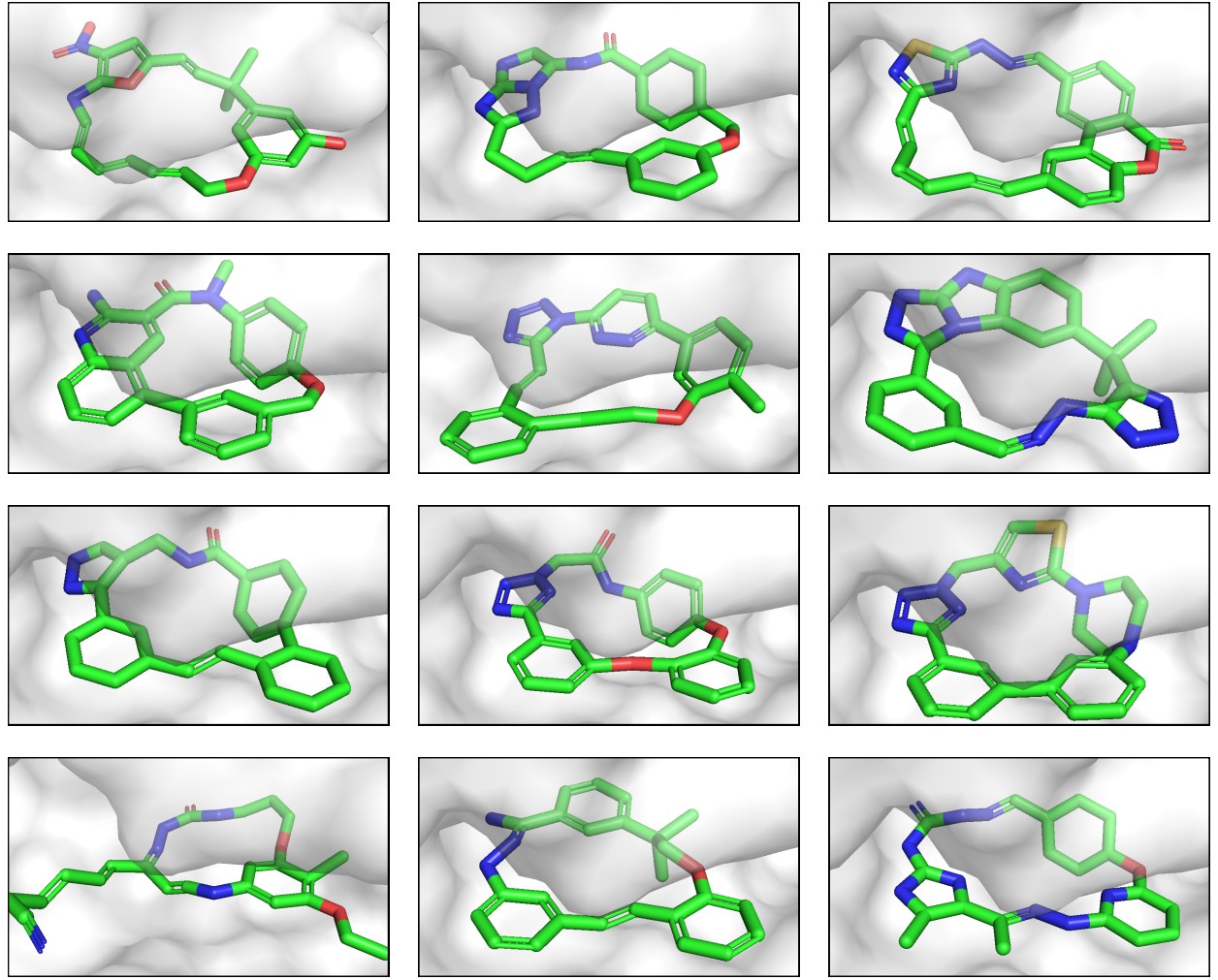

*Figure 17.* **Visualizations of generated molecules in the protein-conditioned setting.** The top parts of molecules are less visible because they are obstructed by a semi-transparent protein pocket.

## J. Additional Results

### J.1. Standard Deviations

In Table 6 we report standard deviations for our experiments. The metric results were computed on 5 sets of 1000 molecules.

*Table 6.* **Standard deviations of unconditional and protein-conditioned macrocycle generation performance.** The last 6 values do not appear for MolDiff as the metrics are defined only in the protein-conditioning setting. Variances for unguided MolDiff and MolSnapper are not reported as generating multiple sets of 1000 macrocycles proved computationally prohibitive due to the very low baseline macrocycle success rates.

| Metrics ($\uparrow$; [0-1]) | MolDiff +MACROGUIDE | MolSnapper +MACROGUIDE |
|---|---|---|
| Validity | $\pm.003$ | $\pm.008$ |
| Connectivity | $\pm.001$ | $\pm.000$ |
| Successfulness | $\pm.003$ | $\pm.008$ |
| Out of successful: | | |
|   Macrocycles | $\pm.002$ | $\pm.003$ |
| Out of macrocycles: | | |
|   Diversity | $\pm.005$ | $\pm.001$ |
|   Novelty | $\pm.000$ | $\pm.001$ |
|   Uniqueness | $\pm.001$ | $\pm.001$ |
|   All PoseBusters tests | - | $\pm.015$ |
|     Ligand PoseBusters | $\pm.013$ | $\pm.018$ |
|       Bond lengths | $\pm.004$ | $\pm.010$ |
|       Bond angles | $\pm.002$ | $\pm.012$ |
|       Internal steric clash | $\pm.013$ | $\pm.017$ |
|       Aromatic ring flatness | $\pm.001$ | $\pm.002$ |
|       Non-ar. ring non-flatness | $\pm.001$ | $\pm.001$ |
|       Double bond flatness | $\pm.004$ | $\pm.002$ |
|       Internal energy | $\pm.004$ | $\pm.010$ |
|     Protein PoseBusters | - | $\pm.008$ |
|       Protein-ligand max. distance | - | $\pm.000$ |
|       Min. distance to protein | - | $\pm.008$ |
|       Volume overlap with protein | - | $\pm.000$ |
|   Pharmacophore satisfaction | - | $\pm.014$ |
|   Macrocycle Lipinski | - | $\pm.017$ |

### J.2. Ablations of Guidance Terms

We perform ablations to show the influence of $H_1$ birth optimization (closing the cycle) and $H_0$ death optimization (keeping the point cloud connected) on macrocycle generation, both in the unconditional and conditional setting. We keep the $H_1$ death optimization unchanged, as it is the core of the method. We generate sets of 1000 molecules with MolDiff+MACROGUIDE and MolSnapper+MACROGUIDE, and report results in Table 7 and Table 8 respectively.

We observe a drop in macrocycle generation when the birth of the $H_1$ component is not constrained as this may lead to the creation of non-closed cycles. On the other hand, the lack of $H_0$ death term, while not detrimental to macrocycle generation performance or macrocycle quality, results in a drop in connected molecules and hence a drop in successfulness (Table 7).

In the protein conditioning setting, the lack of $H_1$ birth term produces a similar drop in performance. However, $H_0$ death optimization can be safely removed, probably because of the confined volume of the protein pocket (Table 8).

### J.3. Ablations of Squares

We perform additional experiments with and without squaring the key equations in the topological loss. The results in Table 9 and Table 10 show that adding a square to $H_1$ birth decreases the rate of macrocycle generation, as well as their PoseBusters performance, both in the unconditional and conditional settings.

Omitting the $H_1$ death square results in a decrease in molecule validity and the PoseBusters checks performance for macrocycles. Finally, the lack of $H_0$ death square seems the least detrimental to model performance, and even improves the performance for some metrics. This term modification can therefore act as a good starting point for adapting MACROGUIDE for specific tasks.

*Table 7.* **Ablations of different guidance terms in the unconditional setting.** The metrics most affected by the ablation are in bold. Some metrics are not reported because not enough successful macrocycles were produced. Results were obtained from sets of 1000 molecules.

| Metrics ($\uparrow$; [0-1]) | MolDiff +MACROGUIDE (ours) | No $H_1$ birth | No $H_0$ | No $H_1$ birth and no $H_0$ |
|---|---|---|---|---|
| Validity | 0.989 | 0.990 | 0.990 | 0.990 |
| Connectivity | 0.999 | 0.954 | **0.863** | **0.022** |
| Successfulness | 0.988 | 0.944 | 0.854 | 0.022 |
| Out of successful: | | | | |
| **Macrocycles** | 0.997 | **0.075** | 0.995 | **0.091** |
| Out of macrocycles: | | | | |
| Diversity | 0.771 | 0.735 | 0.774 | - |
| Novelty | 1.000 | 1.000 | 1.000 | - |
| Uniqueness | 1.000 | 1.000 | 1.000 | - |
| **All PoseBusters tests** | 0.805 | 0.732 | 0.840 | - |
| Bond lengths | 0.990 | 1.000 | 0.994 | - |
| Bond angles | 0.987 | 0.761 | 0.985 | - |
| Internal steric clash | 0.844 | 0.972 | 0.893 | - |
| Aromatic ring flatness | 0.999 | 1.000 | 1.000 | - |
| Non-ar. ring non-flatness | 0.999 | 1.000 | 1.000 | - |
| Double bond flatness | 0.989 | 1.000 | 0.968 | - |
| Internal energy | 0.984 | 1.000 | 0.991 | - |

*Table 8.* **Ablations of different guidance terms in the conditional setting.** The metrics most affected by the ablation are in bold. Some metrics are not reported because not enough successful macrocycles were produced. Results were obtained from sets of 1000 molecules.

| Metrics ($\uparrow$; [0-1]) | MolSnapper +MACROGUIDE (ours) | No $H_1$ birth | No $H_0$ | No $H_1$ birth and no $H_0$ |
|---|---|---|---|---|
| Validity | 0.925 | 0.903 | 0.921 | 0.929 |
| Connectivity | 1.000 | 1.000 | 1.000 | 1.000 |
| Successfulness | 0.925 | 0.903 | 0.921 | 0.929 |
| Out of successful: | | | | |
| **Macrocycles** | 0.995 | **0.078** | 0.989 | **0.016** |
| Out of macrocycles: | | | | |
| Diversity | 0.712 | 0.734 | 0.711 | - |
| Novelty | 1.000 | 1.000 | 1.000 | - |
| Uniqueness | 1.000 | 1.000 | 0.999 | - |
| **All PoseBusters tests** | 0.575 | 0.414 | 0.591 | - |
| Ligand PoseBusters | 0.626 | 0.471 | 0.654 | - |
| Bond lengths | 0.860 | 0.829 | 0.923 | - |
| Bond angles | 0.888 | 0.643 | 0.907 | - |
| Internal steric clash | 0.854 | 0.914 | 0.854 | - |
| Aromatic ring flatness | 0.992 | 0.986 | 0.993 | - |
| Non-ar. ring non-flatness | 0.999 | 1.000 | 1.000 | - |
| Double bond flatness | 0.993 | 0.986 | 0.995 | - |
| Internal energy | 0.921 | 0.929 | 0.926 | - |
| Protein PoseBusters | 0.911 | 0.843 | 0.902 | - |
| Protein-ligand max. distance | 1.000 | 1.000 | 1.000 | - |
| Min. distance to protein | 0.907 | 0.843 | 0.902 | - |
| Volume overlap with protein | 1.000 | 1.000 | 1.000 | - |
| Pharmacophore satisfaction | 0.789 | 0.829 | 0.804 | - |
| Macrocycle Lipinski | 0.638 | 0.743 | 0.580 | - |

## J.4. Guidance Strength

Tables 11 and 12 demonstrate that the chosen guidance strength of 1 performs consistently well across both unconditional and conditional settings.

*Table 9.* **Influence of the square on each guidance term in the unconditional setting.** The metrics most affected by the ablation are in bold. Results were obtained from sets of 1000 molecules.

| Metrics ($\uparrow$; [0-1]) | MolDiff +MACROGUIDE (ours) | +$H_1$ birth square | No $H_1$ death square | No $H_0$ death square |
|---|---|---|---|---|
| Validity | 0.989 | 0.981 | **0.924** | 0.991 |
| Connectivity | 0.999 | 0.985 | 1.000 | 1.000 |
| Successfulness | 0.988 | 0.966 | 0.924 | 0.991 |
| Out of successful: | | | | |
| **Macrocycles** | 0.997 | **0.633** | 0.998 | 1.000 |
| Out of macrocycles: | | | | |
| Diversity | 0.771 | 0.789 | 0.782 | 0.752 |
| Novelty | 1.000 | 1.000 | 1.000 | 1.000 |
| Uniqueness | 1.000 | 1.000 | 1.000 | 1.000 |
| **All PoseBusters tests** | 0.805 | **0.628** | **0.711** | 0.791 |
| Bond lengths | 0.990 | 0.964 | 0.935 | 0.989 |
| Bond angles | 0.987 | 0.702 | 0.938 | 0.987 |
| Internal steric clash | 0.844 | 0.913 | 0.837 | 0.815 |
| Aromatic ring flatness | 0.999 | 0.998 | 0.999 | 0.996 |
| Non-ar. ring non-flatness | 0.999 | 0.998 | 0.999 | 0.996 |
| Double bond flatness | 0.989 | 0.987 | 0.977 | 0.987 |
| Internal energy | 0.984 | 0.959 | 0.953 | 0.987 |

*Table 10.* **Influence of the square on each guidance term in the conditional setting.** The metrics most affected by the ablation are in bold. Results were obtained from sets of 1000 molecules.

| Metrics ($\uparrow$; [0-1]) | MolSnapper +MACROGUIDE (ours) | +$H_1$ birth square | No $H_1$ death square | No $H_0$ death square |
|---|---|---|---|---|
| Validity | 0.925 | 0.914 | **0.883** | 0.915 |
| Connectivity | 1.000 | 1.000 | 1.000 | 1.000 |
| Successfulness | 0.925 | 0.914 | 0.883 | 0.915 |
| Out of successful: | | | | |
| **Macrocycles** | 0.995 | **0.521** | 0.998 | 0.991 |
| Out of macrocycles: | | | | |
| Diversity | 0.712 | 0.711 | 0.717 | 0.703 |
| Novelty | 1.000 | 1.000 | 1.000 | 1.000 |
| Uniqueness | 1.000 | 1.000 | 1.000 | 1.000 |
| **All PoseBusters tests** | 0.575 | **0.309** | 0.529 | 0.572 |
| Ligand PoseBusters | 0.626 | 0.332 | 0.580 | 0.607 |
| Bond lengths | 0.860 | 0.800 | 0.815 | 0.869 |
| Bond angles | 0.888 | 0.502 | 0.879 | 0.905 |
| Internal steric clash | 0.854 | 0.851 | 0.846 | 0.847 |
| Aromatic ring flatness | 0.992 | 0.987 | 0.993 | 0.981 |
| Non-ar. ring non-flatness | 0.999 | 1.000 | 1.000 | 1.000 |
| Double bond flatness | 0.993 | 0.989 | 0.992 | 0.999 |
| Internal energy | 0.921 | 0.905 | 0.910 | 0.899 |
| Protein PoseBusters | 0.911 | 0.891 | 0.915 | 0.940 |
| Protein-ligand max. distance | 1.000 | 1.000 | 1.000 | 1.000 |
| Min. distance to protein | 0.907 | 0.891 | 0.915 | 0.940 |
| Volume overlap with protein | 1.000 | 1.000 | 0.999 | 1.000 |
| Pharmacophore satisfaction | 0.789 | 0.824 | 0.820 | 0.744 |
| Macrocycle Lipinski | 0.638 | 0.670 | 0.640 | 0.630 |

*Table 11.* **Influence of guidance strength on performance in the unconditional setting.** The original results from Table 3 correspond to the strength of 1. Results were obtained from 1000 samples.

| Metrics (↑; [0-1]) | 0.1 | 0.5 | MolDiff +MACROGUIDE 1.0 (ours) | 1.5 | 2.0 |
|---|---|---|---|---|---|
| Validity | 0.994 | 0.995 | 0.989 | 0.991 | 0.992 |
| Connectivity | 0.999 | 0.996 | 0.999 | 0.996 | 0.988 |
| Successfulness | 0.993 | 0.991 | 0.988 | 0.987 | 0.980 |
| Out of successful: | | | | | |
| **Macrocycles** | 0.769 | 0.967 | 0.997 | 0.990 | 0.754 |
| Out of macrocycles: | | | | | |
| Diversity | 0.752 | 0.763 | 0.771 | 0.780 | 0.796 |
| Novelty | 1.000 | 1.000 | 1.000 | 1.000 | 1.000 |
| Uniqueness | 1.000 | 0.999 | 1.000 | 0.992 | 1.000 |
| **All PoseBusters tests** | 0.662 | 0.754 | 0.805 | 0.870 | 0.548 |
| Bond lengths | 0.986 | 0.996 | 0.990 | 0.997 | 0.961 |
| Bond angles | 0.808 | 0.947 | 0.987 | 0.939 | 0.608 |
| Internal steric clash | 0.823 | 0.813 | 0.844 | 0.960 | 0.963 |
| Aromatic ring flatness | 0.992 | 0.997 | 0.999 | 1.000 | 1.000 |
| Non-ar. ring non-flatness | 0.995 | 0.997 | 0.999 | 0.999 | 1.000 |
| Double bond flatness | 0.986 | 0.990 | 0.989 | 0.990 | 0.991 |
| Internal energy | 0.976 | 0.977 | 0.984 | 0.975 | 0.951 |

*Table 12.* **Influence of guidance strength on performance in the conditional setting.** The original results from Table 4 correspond to the strength of 1. Results were obtained from 1000 samples.

| Metrics (↑; [0-1]) | 0.1 | 0.5 | MolSnapper +MACROGUIDE 1.0 (ours) | 1.5 | 2.0 |
|---|---|---|---|---|---|
| Validity | 0.924 | 0.933 | 0.925 | 0.876 | 0.891 |
| Connectivity | 1.000 | 1.000 | 1.000 | 1.000 | 1.000 |
| Successfulness | 0.924 | 0.933 | 0.925 | 0.876 | 0.891 |
| Out of successful: | | | | | |
| **Macrocycles** | 0.411 | 0.867 | 0.995 | 0.995 | 0.845 |
| Out of macrocycles: | | | | | |
| Diversity | 0.655 | 0.695 | 0.712 | 0.724 | 0.724 |
| Novelty | 1.000 | 1.000 | 1.000 | 1.000 | 1.000 |
| Uniqueness | 1.000 | 1.000 | 1.000 | 1.000 | 1.000 |
| **All PoseBusters tests** | 0.384 | 0.507 | 0.575 | 0.495 | 0.154 |
| Ligand PoseBusters | 0.426 | 0.561 | 0.626 | 0.561 | 0.206 |
| Bond lengths | 0.876 | 0.890 | 0.860 | 0.854 | 0.789 |
| Bond angles | 0.658 | 0.791 | 0.888 | 0.802 | 0.311 |
| Internal steric clash | 0.824 | 0.847 | 0.854 | 0.874 | 0.870 |
| Aromatic ring flatness | 0.995 | 0.994 | 0.992 | 0.987 | 0.991 |
| Non-ar. ring non-flatness | 1.000 | 0.999 | 0.999 | 0.999 | 0.999 |
| Double bond flatness | 0.982 | 0.984 | 0.993 | 0.993 | 0.985 |
| Internal energy | 0.876 | 0.927 | 0.921 | 0.911 | 0.902 |
| Protein PoseBusters | 0.895 | 0.886 | 0.911 | 0.857 | 0.744 |
| Protein-ligand max. distance | 1.000 | 1.000 | 1.000 | 1.000 | 1.000 |
| Min. distance to protein | 0.895 | 0.886 | 0.907 | 0.857 | 0.744 |
| Volume overlap with protein | 1.000 | 1.000 | 1.000 | 1.000 | 0.993 |
| Pharmacophore satisfaction | 0.800 | 0.808 | 0.789 | 0.828 | 0.838 |
| Macrocycle Lipinski | 0.603 | 0.642 | 0.638 | 0.673 | 0.659 |

### J.5. Vietoris-Rips vs. Alpha Complex

Alpha complexes are computationally more efficient than Vietoris-Rips (VR) complexes, and faster to compute in isolation on molecule-sized point clouds. We considered them during development but ultimately selected VR for three reasons.

**1. Empirical performance.** End-to-end sampling experiments showed that Alpha guidance is strictly worse across all metrics: validity 85% vs. 99%, macrocycle rate 94% vs. 100%. Conceptually, VR applies a direct repulsive force across the shortest cross-ring chord, naturally keeping the macrocycle cavity open. Alpha optimizes a geometric circumradius, which can inadvertently distort local covalent bonds to satisfy the topological penalty.

**2. Metric flexibility.** VR requires only a pairwise distance matrix and therefore accommodates non-Euclidean metrics, for instance, distances derived from bond-prediction logits. Alpha complexes require actual Euclidean coordinates in $\mathbb{R}^3$ to construct the underlying Delaunay triangulation, and cannot operate on arbitrary distance matrices. This flexibility in VR leaves open the possibility of integrating bond-level information in future work.

**3. Negligible runtime overhead.** VR computation accounts for only $\sim 8\%$ of total sampling time (the denoiser dominates), reducible to $\sim 2\%$ with $k$-step guidance (Section J.9.1). The speed advantage of Alpha is therefore negligible in the full pipeline.

### J.6. Base Model Dependency

To assess how MACROGUIDE performs when the underlying generative model is degraded, we inject Gaussian noise into the pretrained MolDiff weights at increasing scales relative to each parameter's standard deviation. Table 13 shows that MACROGUIDE achieves a 100% macrocycle rate among connected molecules at all viable noise levels (up to 1% relative noise), even as base validity and connectivity degrade. This confirms that MACROGUIDE decouples global topology from local chemistry: it robustly enforces macrocyclic topology while relying on the base model only for chemical validity. At 2% noise the model collapses entirely and no connected molecules are produced regardless of guidance, establishing the hard floor set by base model quality.

*Table 13.* **MACROGUIDE with degraded base models.** Gaussian noise is injected into MolDiff's pretrained weights at increasing scales (relative to each parameter's standard deviation). Results are obtained from 20 molecules per condition. A dash indicates that no connected molecules were produced.

| Noise scale | No guidance | | | MACROGUIDE | | |
|---|---|---|---|---|---|---|
| | Valid. | Conn. | Macro. | Valid. | Conn. | Macro. |
| 0.000 | 0.95 | 0.90 | 0.11 | 0.95 | 0.95 | **1.00** |
| 0.001 | 0.85 | 0.80 | 0.19 | 0.85 | 0.85 | **1.00** |
| 0.002 | 0.80 | 0.80 | 0.19 | 0.75 | 0.75 | **1.00** |
| 0.005 | 0.80 | 0.75 | 0.13 | 0.65 | 0.65 | **1.00** |
| 0.010 | 0.80 | 0.80 | 0.06 | 0.70 | 0.70 | **1.00** |
| 0.020 | 0.15 | 0.10 | 0.00 | 0.00 | 0.00 | — |

### J.7. Atom Type Distribution

Table 14 reports the distribution of atom types in the generated macrocycles.

*Table 14.* **Atom distribution of generated macrocycles.** Represented as a fraction of all atoms.

| Method | C | N | O | F | P | S | Cl |
|---|---|---|---|---|---|---|---|
| MolDiff (no guid.) | 0.7978 | 0.0798 | 0.1089 | 0.0011 | 0.0000 | 0.0113 | 0.0011 |
| +MACROGUIDE (ours) | 0.8785 | 0.0477 | 0.0722 | 0.0001 | 0.0000 | 0.0014 | 0.0001 |
| MolSnapper (no guid.) | 0.7508 | 0.1749 | 0.0714 | 0.0002 | 0.0000 | 0.0024 | 0.0003 |
| +MACROGUIDE (ours) | 0.7540 | 0.1921 | 0.0483 | 0.0000 | 0.0000 | 0.0051 | 0.0005 |

### J.8. Ring Size Distribution

Table 15 reports the number of small rings present in the generated molecules. Generated macrocycles typically contain a few additional rings, indicating that they are not merely simple large cycles but exhibit nontrivial structural complexity.

Moreover, MACROGUIDE reduces the number of 3- and 4-membered rings, which are the most strained and therefore undesirable, leading to chemically more favourable structures.

*Table 15.* **Ring size distribution in generated macrocycles.** Represented as the average number of cycles of a certain size per molecule.

| Method | 3-cycles ($\downarrow$) | 4-cycles ($\downarrow$) | 5-cycles | 6-cycles | 7-cycles | 8-cycles | 9-cycles |
|---|---|---|---|---|---|---|---|
| MolDiff (no guid.) | 0.054 | 0.022 | 0.482 | 2.552 | 0.154 | 0.022 | 1.885 |
| +MACROGUIDE (ours) | 0.065 | 0.022 | 0.449 | 1.933 | 0.033 | 0.003 | 1.229 |
| MolSnapper (no guid.) | 0.258 | 0.082 | 1.005 | 2.919 | 0.133 | 0.009 | 2.216 |
| +MACROGUIDE (ours) | 0.183 | 0.077 | 1.123 | 2.127 | 0.039 | 0.001 | 1.341 |

## J.9. Runtime Comparison

### J.9.1. APPLYING GUIDANCE EVERY FEW STEPS

To reduce computational cost, the topological guidance can be applied only every few denoising iterations. Table 16 shows that applying MACROGUIDE once every up to 5 steps results in only a slight degradation of the performances, while substantially decreasing the overall computational overhead.

*Table 16.* **Performance of protein-conditioned macrocycle generation with guidance applied every few steps.** The first two columns correspond to Table 4. Values outperforming or matching the baseline MolSnapper metrics are in bold.

| Metrics ($\uparrow$; [0-1]) | MolSnapper (no guid.) | Every 1 | Every 2 | Every 3 | Every 4 | Every 5 |
|---|---|---|---|---|---|---|
| Validity | 0.858 | **0.925** | **0.930** | **0.942** | **0.914** | **0.940** |
| Connectivity | 1.000 | **1.000** | **1.000** | **1.000** | **1.000** | **1.000** |
| Successfulness | 0.858 | **0.925** | **0.930** | **0.942** | **0.914** | **0.940** |
| Out of successful: | | | | | | |
| **Macrocycles** | 0.003 | **0.995** | **0.943** | **0.911** | **0.864** | **0.852** |
| Out of macrocycles: | | | | | | |
| Diversity | 0.626 | **0.712** | **0.709** | **0.713** | **0.711** | **0.723** |
| Novelty | 1.000 | **1.000** | **1.000** | **1.000** | **1.000** | **1.000** |
| Uniqueness | 1.000 | **1.000** | **1.000** | **1.000** | **1.000** | **1.000** |
| **All PoseBusters tests** | 0.440 | **0.575** | **0.590** | **0.569** | **0.535** | **0.503** |
| Ligand PoseBusters | 0.539 | **0.626** | **0.658** | **0.661** | **0.637** | **0.624** |
| Bond lengths | 0.844 | **0.860** | **0.900** | **0.927** | **0.925** | **0.928** |
| Bond angles | 0.913 | 0.888 | **0.918** | **0.916** | 0.890 | 0.850 |
| Internal steric clash | 0.862 | 0.854 | 0.856 | 0.851 | 0.839 | **0.869** |
| Aromatic ring flatness | 0.996 | 0.992 | 0.995 | 0.994 | **0.997** | 0.990 |
| Non-ar. ring non-flatness | 0.993 | **0.999** | **1.000** | **0.999** | **0.999** | **1.000** |
| Double bond flatness | 0.980 | **0.993** | **0.992** | **0.986** | **0.985** | **0.983** |
| Internal energy | 0.818 | **0.921** | **0.928** | **0.920** | **0.916** | **0.920** |
| Protein PoseBusters | 0.806 | **0.911** | **0.895** | **0.875** | **0.838** | 0.792 |
| Protein-ligand max. distance | 1.000 | **1.000** | **1.000** | **1.000** | **1.000** | **1.000** |
| Min. distance to protein | 0.806 | **0.907** | **0.895** | **0.875** | **0.838** | 0.792 |
| Volume overlap with protein | 1.000 | **1.000** | **1.000** | **1.000** | **1.000** | 0.999 |
| Pharmacophore satisfaction | 0.769 | **0.789** | **0.791** | **0.800** | **0.820** | **0.823** |
| Macrocycle Lipinski | 0.551 | **0.638** | **0.692** | **0.698** | **0.682** | **0.693** |

### J.9.2. STARTING TOPOLOGICAL GUIDANCE LATE

Table 17 analyzes the effect of delaying the onset of topological guidance. The guidance can be introduced halfway through the denoising process with minimal impact on molecular validity, macrocycle-related metrics, and PoseBusters performance.

## J.10. Flow Matching Models

Whereas diffusion models sample by reversing a stochastic noising process guided by a learned score, flow matching learns a deterministic velocity field that continuously transports noise to data along smooth paths (Lipman et al., 2022). To demonstrate the generality of MACROGUIDE, we apply it to FLOWR.root (Cremer et al., 2025), a state-of-the-art flow-matching model for molecules that supports unconditional and protein-conditioned generation, among other modes. Although MACROGUIDE was originally designed for diffusion, we incorporate it directly into the velocity field, preserving

*Table 17.* **Performance of protein-conditioned macrocycle generation with guidance starting late.** The number denotes the number of remaining denoising steps when the guidance is applied, out of 1000 steps total. The first two columns correspond to Table 4. Values outperforming or matching the baseline MolSnapper metrics are in bold. Non-baseline results were obtained from sets of 200 molecules.

| Metrics (↑; [0-1]) | MolSnapper (no guid.) | 1000 steps | 900 steps | 800 steps | 700 steps | 600 steps | 500 steps | 400 steps | 300 steps | 200 steps | 100 steps |
|---|---|---|---|---|---|---|---|---|---|---|---|
| Validity | 0.858 | **0.925** | **0.925** | **0.925** | **0.910** | **0.920** | **0.940** | **0.905** | **0.910** | **0.860** | 0.840 |
| Connectivity | 1.000 | **1.000** | **1.000** | **1.000** | **1.000** | **1.000** | **1.000** | **1.000** | **1.000** | **1.000** | **1.000** |
| Successfulness | 0.858 | **0.925** | **0.925** | **0.925** | **0.910** | **0.920** | **0.940** | **0.905** | **0.910** | **0.860** | 0.840 |
| Out of successful: | | | | | | | | | | | |
| **Macrocycles** | 0.003 | **0.995** | **0.995** | **0.989** | **0.984** | **0.995** | **0.995** | **1.000** | **0.995** | **0.965** | **0.935** |
| Out of macrocycles: | | | | | | | | | | | |
| Diversity | 0.626 | **0.712** | **0.711** | **0.698** | **0.695** | **0.672** | **0.686** | **0.676** | **0.670** | **0.672** | **0.704** |
| Novelty | 1.000 | **1.000** | **1.000** | **1.000** | **1.000** | **1.000** | **1.000** | **1.000** | **1.000** | **1.000** | **1.000** |
| Uniqueness | 1.000 | **1.000** | **1.000** | **1.000** | **1.000** | **1.000** | **1.000** | **1.000** | **1.000** | **1.000** | **1.000** |
| **All PoseBusters tests** | 0.440 | **0.575** | **0.598** | **0.552** | **0.547** | **0.503** | **0.551** | **0.486** | 0.376 | 0.319 | 0.178 |
| Ligand PoseBusters | 0.539 | **0.626** | **0.658** | **0.623** | **0.654** | **0.601** | **0.684** | **0.652** | **0.547** | 0.446 | 0.306 |
| Bond lengths | 0.844 | **0.860** | **0.880** | **0.869** | **0.922** | **0.863** | **0.914** | **0.884** | **0.917** | **0.886** | 0.834 |
| Bond angles | 0.913 | 0.888 | 0.902 | 0.852 | **0.927** | 0.902 | 0.888 | **0.923** | 0.834 | 0.789 | 0.561 |
| Internal steric clash | 0.862 | 0.854 | **0.864** | **0.896** | 0.855 | 0.814 | **0.877** | **0.873** | **0.890** | 0.813 | 0.841 |
| Aromatic ring flatness | 0.996 | 0.992 | **1.000** | **1.000** | 0.994 | **1.000** | 0.995 | 0.994 | 0.994 | **1.000** | **1.000** |
| Non-ar. ring non-flatness | 0.993 | **0.999** | **1.000** | **1.000** | **1.000** | **1.000** | 0.995 | **1.000** | **1.000** | **1.000** | **1.000** |
| Double bond flatness | 0.980 | **0.993** | **0.995** | **0.989** | **0.994** | **0.995** | **0.989** | **0.989** | 0.961 | 0.910 | 0.930 |
| Internal energy | 0.818 | **0.921** | **0.929** | **0.951** | **0.894** | **0.891** | **0.925** | **0.923** | 0.796 | **0.867** | **0.822** |
| Protein PoseBusters | 0.806 | **0.911** | **0.913** | **0.902** | **0.844** | **0.852** | 0.786 | 0.790 | 0.740 | 0.765 | 0.561 |
| Protein-ligand max. distance | 1.000 | **1.000** | **1.000** | **1.000** | **1.000** | **1.000** | **1.000** | **1.000** | **1.000** | **1.000** | **1.000** |
| Min. distance to protein | 0.806 | **0.907** | **0.913** | **0.902** | **0.844** | **0.852** | 0.786 | 0.790 | 0.740 | 0.765 | 0.561 |
| Volume overlap with protein | 1.000 | **1.000** | **1.000** | **1.000** | **1.000** | **1.000** | **1.000** | **1.000** | **1.000** | **1.000** | **1.000** |
| Pharmacophore satisfaction | 0.769 | **0.789** | **0.804** | **0.809** | **0.816** | **0.820** | **0.893** | **0.823** | **0.818** | **0.819** | **0.815** |
| Macrocycle Lipinski | 0.551 | **0.638** | **0.598** | **0.639** | **0.598** | **0.557** | **0.620** | **0.602** | **0.713** | **0.669** | **0.732** |

*Table 18.* **Performance of protein-conditioned macrocycle generation in a flow matching setup.**

| | FLOWR.root | +MACROGUIDE |
|---|---|---|
| Validity | 0.96 | 0.91 |
| Macrocycle rate (out of valid) | 0.04 | **0.98** |

key advantages of flow-based methods such as fast sampling. Experiments were conducted on the BACE protein pocket using the provided setup and the v.2.1 checkpoint. As shown in Table 18, MACROGUIDE raises the macrocycle rate among valid molecules from 0.04 to 0.98, with only a slight decrease in validity. This minor drop may stem in part from the deterministic velocity field, which lacks the stochastic correction that keeps diffusion samples on the data manifold. However, it may also reflect that the publicly released checkpoint we use was not the fully converged model.

Nonetheless, these results show that MACROGUIDE transfers effectively to flow matching, achieving near-complete macrocyclization while retaining fast sampling. We see this as a promising direction and believe that adapting the guidance more carefully to the deterministic dynamics of flow matching, and testing it across more flow-matching models will close the remaining gap.

# K. Competing Forces Under Protein Conditioning

We provide a detailed analysis of why finetuning collapses under protein conditioning (dropping to an $18\%$ macrocycle rate) while MACROGUIDE remains unaffected ($99\%$), despite neither method being trained on protein-conditioned macrocyclic data. The key distinction lies in the *speed and robustness* of topological enforcement: MACROGUIDE converges to macrocyclic topology within $\sim 100$ steps and resists perturbations, while finetuning encodes only a slow, fragile implicit bias that is easily overridden by protein-pocket constraints.

**Protein conditioning suppresses ring size.** Figure 18 shows the distribution of the largest chordless ring size across methods. In the unconditional case, finetuning successfully shifts the ring size distribution above the macrocycle threshold of 12 atoms. Under protein conditioning, however, finetuning produces a large fraction of molecules with ring sizes 9–11, just below the threshold. The protein pocket's spatial and pharmacophore constraints thus act as a competing force that decreases cycle size. MACROGUIDE, by contrast, maintains a clean distribution concentrated at ring sizes 12–16 in both settings.

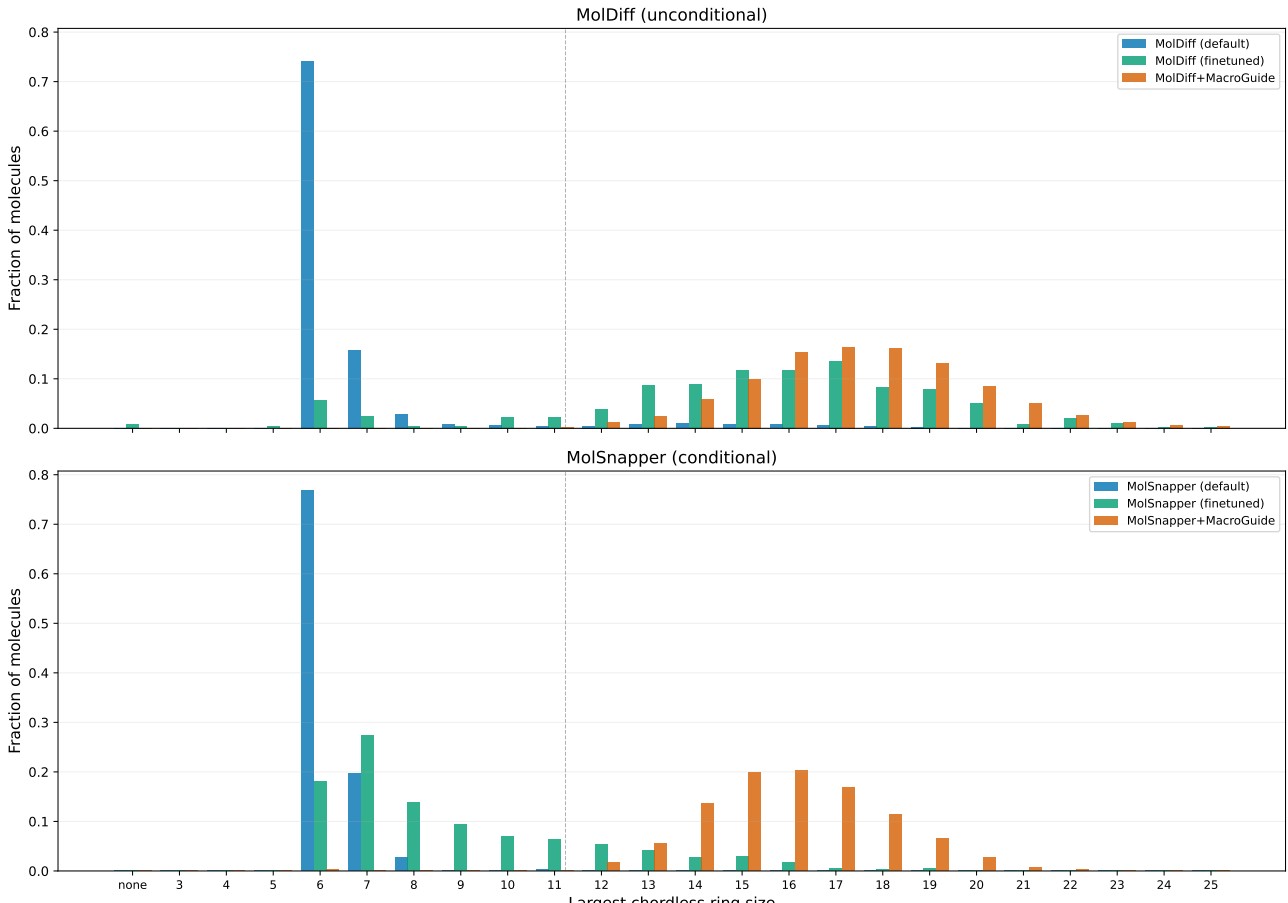

*Figure 18.* **Distribution of largest chordless ring size.** *Top:* Unconditional generation. Both the finetuned model and MACROGUIDE produce a substantially higher proportion of large cycles compared to the default method. *Bottom:* Protein-conditioned generation. Both produce smaller cycles on average; however, MACROGUIDE maintains ring sizes within the macrocyclic regime, while many finetuned samples fall short of the macrocycle threshold by 1–3 atoms.

**Two competing forces.** The generation process under protein conditioning can be understood as balancing two objectives: (i) *macrocyclicity* (forming a large ring) and (ii) *pocket compatibility* (satisfying pharmacophore constraints and avoiding steric clashes). These forces compete: the pocket constraints favor compact geometries and specific atom placements, while macrocyclicity requires a globally extended ring. For the finetuned model, macrocyclicity is encoded only as an implicit prior learned during finetuning. This prior is weak: the $H_1$ death feature drifts slowly and noisily toward the macrocyclic range over the entire 1000-step denoising trajectory. For MACROGUIDE, both objectives are explicitly encoded

as concurrent forces; the topological guidance locks in the macrocyclic ring within the first ∼100–200 steps, and since this enforcement is rapid and applied at every step, it cannot be overridden by the competing pocket forces.

**Perturbation robustness.** To probe the strength of each macrocyclicity prior, we apply a mid-trajectory perturbation: at denoising step 500, atomic positions are perturbed with Gaussian noise ($\sigma = 1.0$ Å), and we track recovery. Figures 19 and 20 show the unconditional setting; Figures 21 and 22 show the conditional setting. MACROGUIDE recovers fully after perturbation (converging $H_1$ death back into the target band), while the finetuned model does not. This confirms that the base model's chemical validity prior is robust to perturbation, but the macrocyclic prior learned through finetuning is not. Under conditioning, the failure mode is more severe (fragmentation rather than mere loss of ring), consistent with pocket constraints acting as a persistent perturbation throughout denoising.

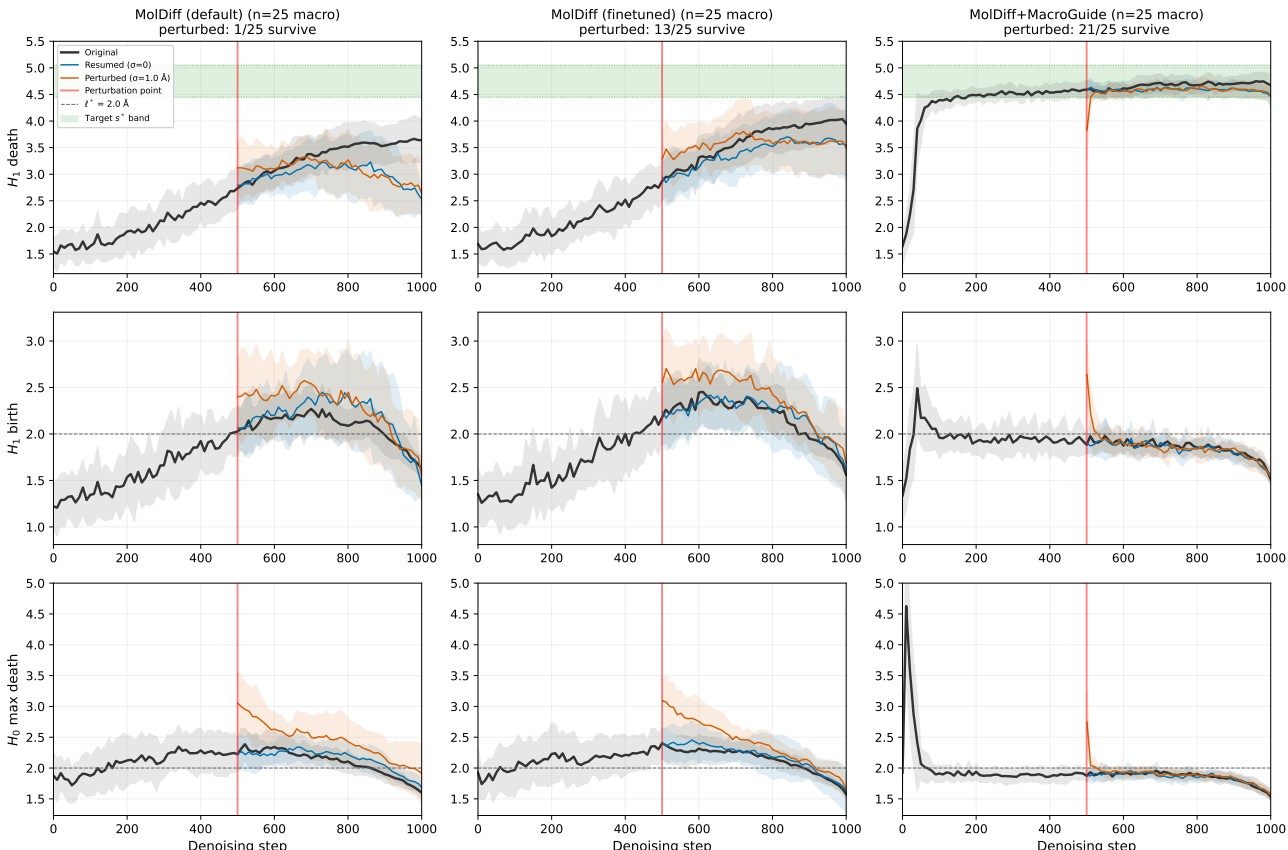

*Figure 19.* **Averaged topological response to mid-trajectory perturbation (unconditional).** Persistent homology features averaged over $n = 25$ originally macrocyclic trajectories per method (shaded regions show one standard deviation). At step 500, positions are perturbed with Gaussian noise ($\sigma = 1.0$ Å). Macrocycle survival after perturbation: default 1/25, finetuned 13/25, MACROGUIDE 21/25. MACROGUIDE converges $H_1$ death into the target band within ∼100 steps and recovers fully, while the finetuned model drifts slowly and noisily toward macrocyclic topology over the entire trajectory.

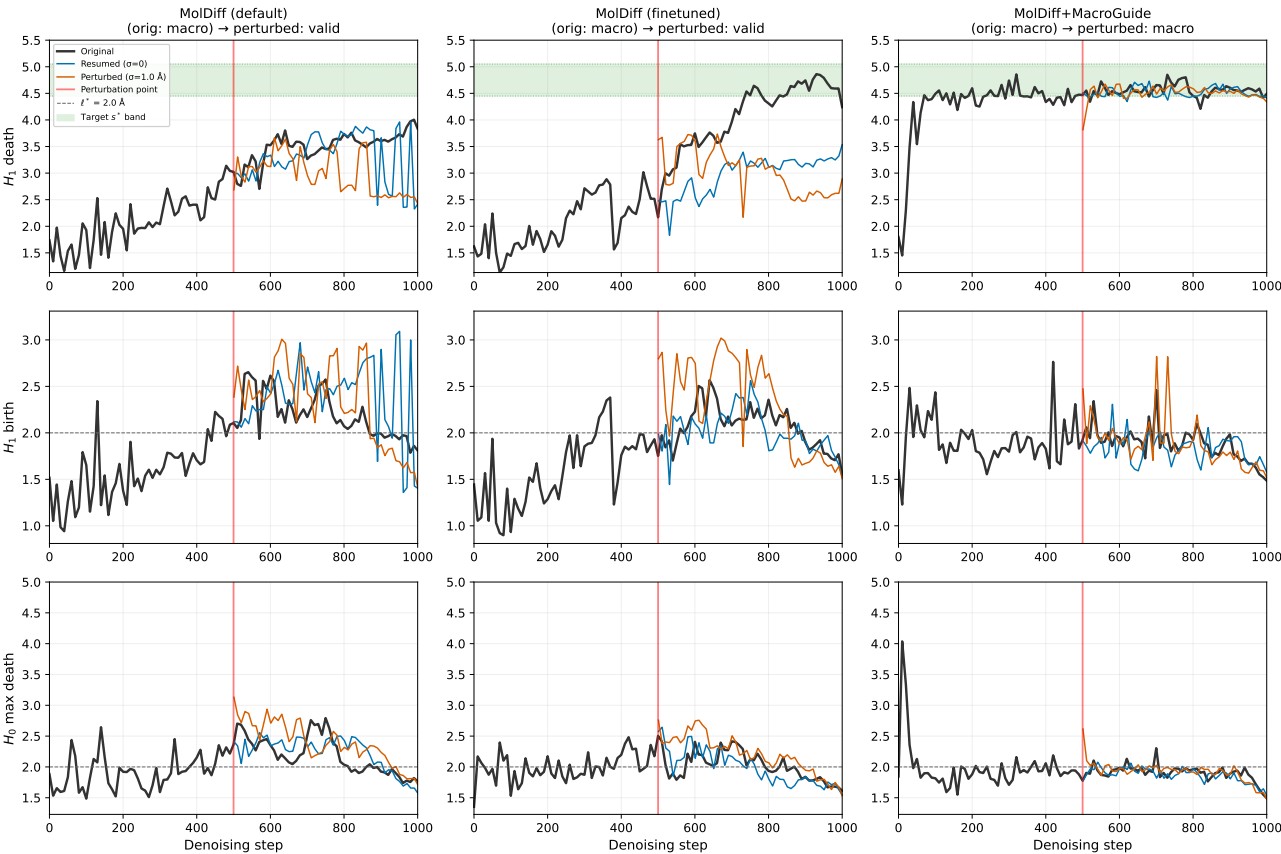

*Figure 20.* **Topological robustness of individual macrocyclic trajectories (unconditional).** Denoising trajectories of three individual MolDiff-generated macrocycles, tracked via persistent homology features ($H_1$ death, $H_1$ birth, $H_0$ max death). At step 500, positions are perturbed with Gaussian noise ($\sigma = 1.0$ Å). The green band marks the target $H_1$ death range $s^* = [4.45, 5.05]$ Å. Only MACROGUIDE recovers a macrocycle after perturbation; both the default and finetuned models yield valid but non-macrocyclic molecules.

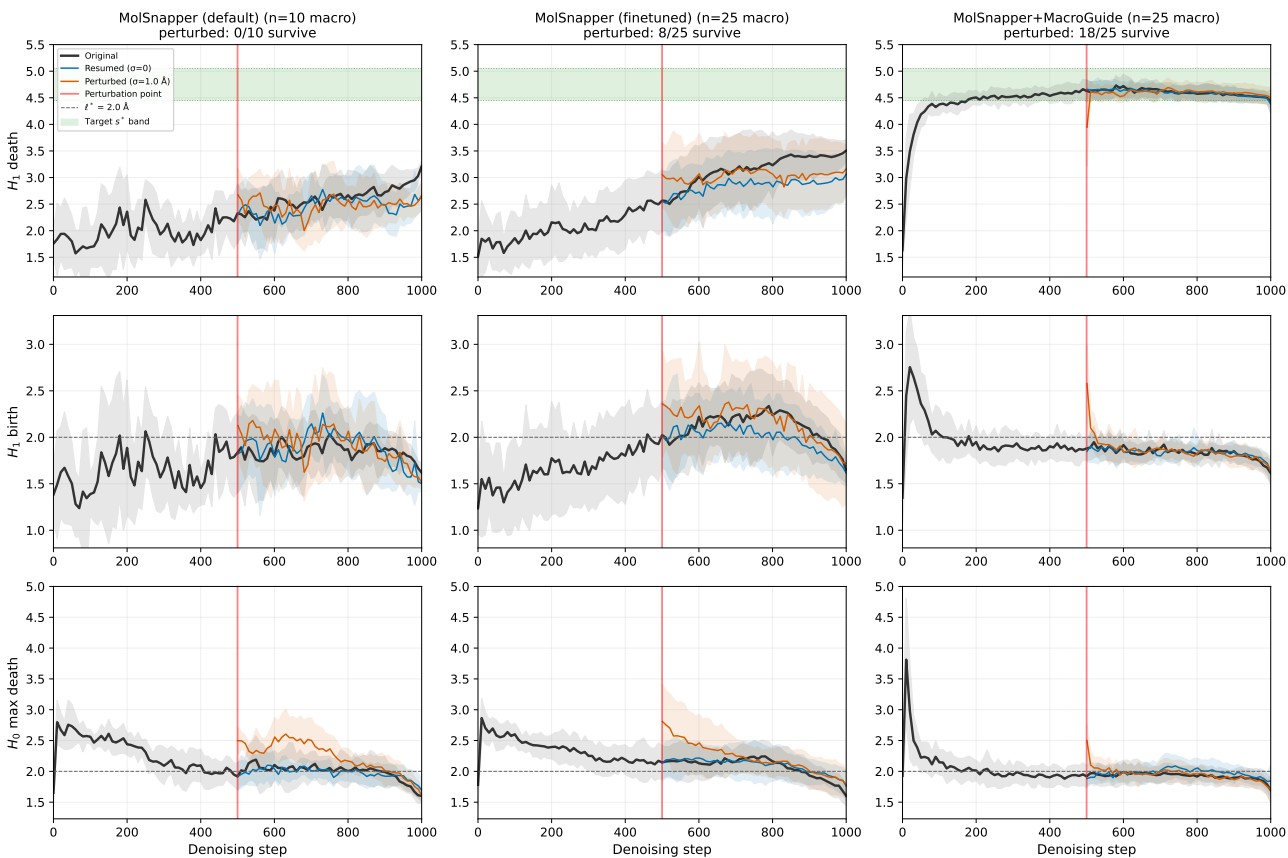

*Figure 21.* **Averaged topological response to mid-trajectory perturbation (conditional).** Persistent homology features averaged over originally macrocyclic trajectories under protein-pocket conditioning (shaded regions show one standard deviation). At step 500, positions are perturbed with Gaussian noise ($\sigma = 1.0$ Å). Available conditional macrocycle trajectories: default $n = 10$, finetuned $n = 25$, MACROGUIDE $n = 25$. Macrocycle survival after perturbation: default $0/10$, finetuned $8/25$, MACROGUIDE $18/25$. Because the finetuned model's implicit bias is too slow to establish ring topology before pocket constraints take effect, the pocket-compatibility objective dominates.

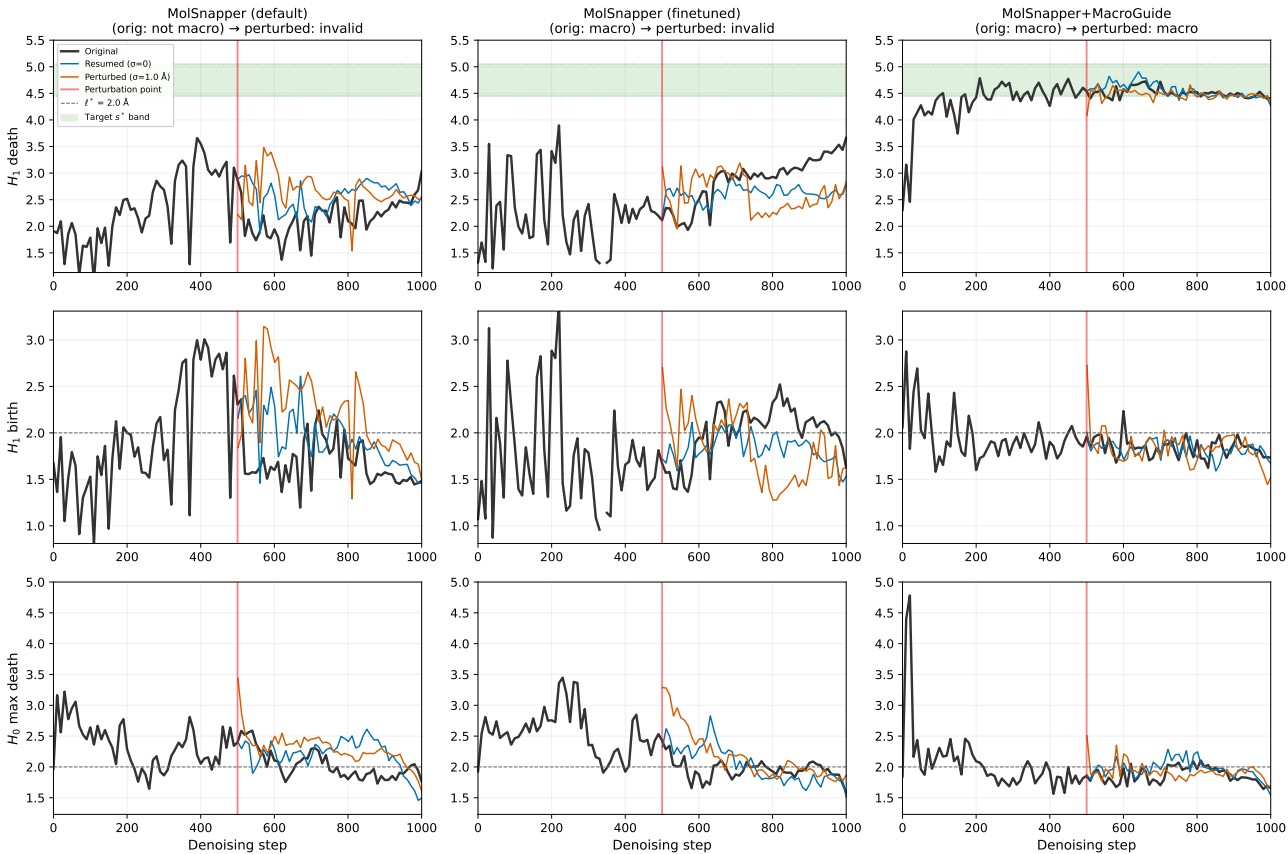

*Figure 22.* **Topological robustness of individual macrocyclic trajectories (conditional).** Denoising trajectories of three individual MolSnapper-generated molecules under protein-pocket conditioning. At step 500, positions are perturbed with Gaussian noise ($\sigma = 1.0$ Å). The default model does not produce a conditional macrocycle even without perturbation. The finetuned model's macrocycle fragments upon perturbation, while MACROGUIDE recovers.

**MACROGUIDE does not force macrocycles where they do not fit.** Importantly, MACROGUIDE maintains a tight pocket fit without compromising compatibility. Figure 23 shows that the mean minimum distance to the pocket, the fraction of atoms within 4 Å of the pocket, and the steric clash rate all converge to virtually identical values across methods. Figure 24 shows that the minimum distance to the deepest residue (GLU81) also remains the same, while the centroid distance increases: MACROGUIDE macrocycles extend further outward from the pocket center to better utilize the available space and accommodate a larger ring. Figure 25 visualizes this behavior for a representative molecule.

**Feature space.** A t-SNE embedding based on topological features ($H_1$ death, $H_1$ birth, $H_0$ death) confirms this analysis (Figure 26). In the unconditional setting, both methods produce macrocycles that align with the GEOM-Drug and Macrocycle-DB macrocycle clusters. Under protein conditioning, however, the finetuned model's outputs collapse toward the default MolSnapper distribution, while MACROGUIDE preserves the manifold structure of the unconditional case.

**Summary.** The key advantage of MACROGUIDE is not merely that it optimizes topology directly, but that it does so *fast and robustly* (converging in ∼100 steps and recovering from perturbations) whereas finetuning produces only a slow, fragile implicit bias that is easily overridden by protein-conditioning forces.

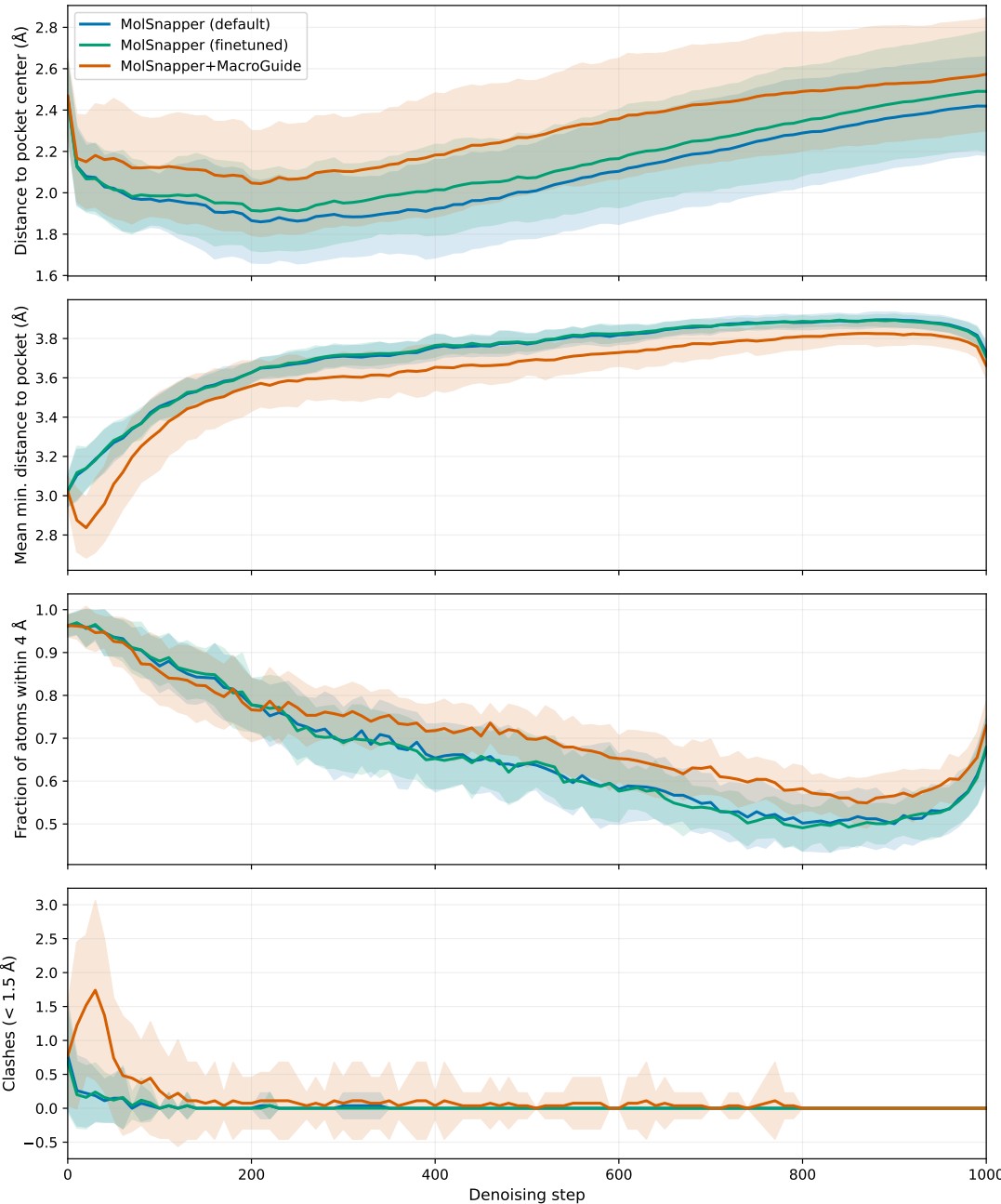

*Figure 23.* **Pocket fit metrics.** (a) Distance from the molecule centroid to the pocket center: MACROGUIDE shows slightly higher distances, indicating it extends outward to better utilize available space. (b) Mean minimum distance from each generated atom to the nearest pocket atom: all methods are similar. (c) Fraction of atoms within 4 Å of any pocket atom: MACROGUIDE achieves slightly better coverage. (d) Number of steric clashes: MACROGUIDE maintains macrocyclic topology without compromising proximity to the pocket.

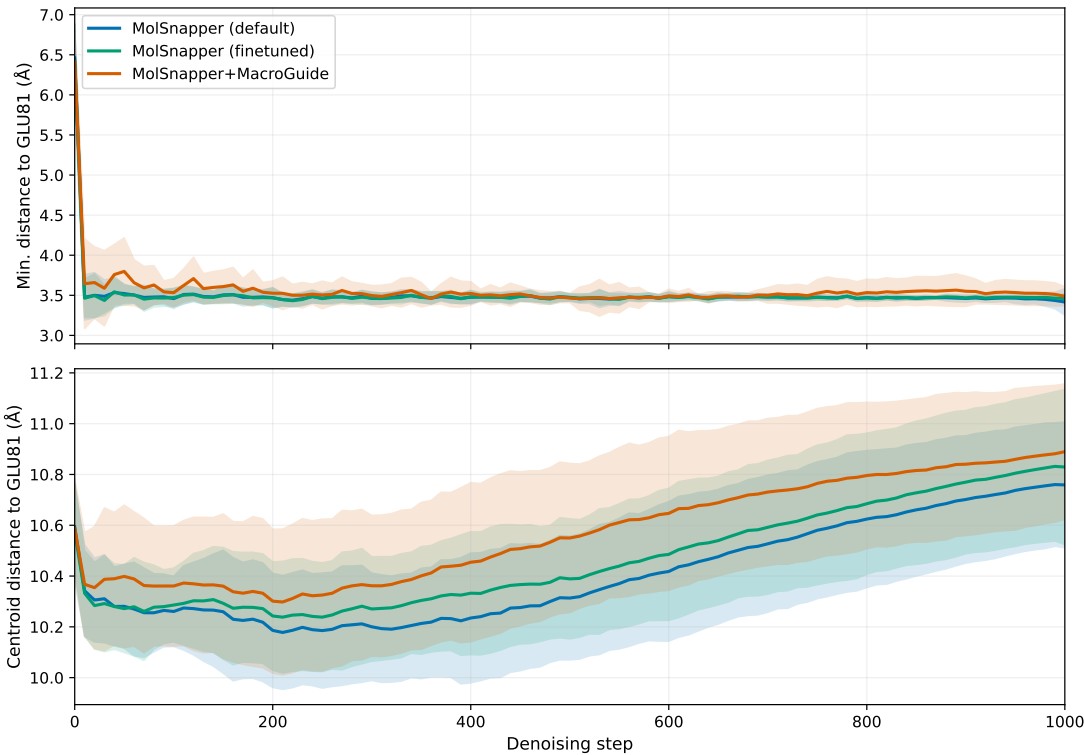

*Figure 24.* **Distance to GLU81, the deepest residue in the protein pocket.** MACROGUIDE preserves a similar minimum distance to the residue, indicating molecules continue to fit tightly within the pocket. The centroid distance increases, suggesting MACROGUIDE macrocycles extend further out of the pocket, better utilizing available space and accommodating larger rings.

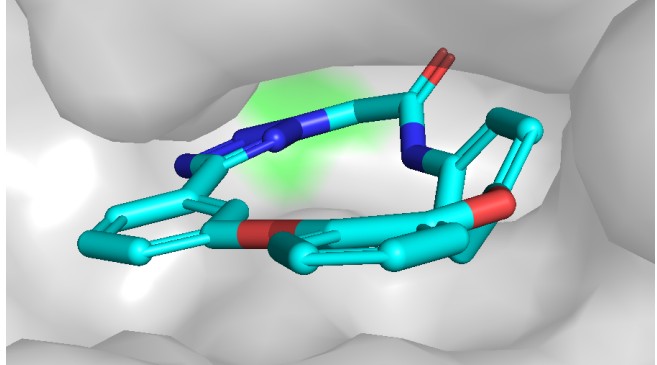

*Figure 25.* **Visualization of the GLU81 residue.** The residue is shown in green and serves as a reference for how deeply the molecule sits within the protein pocket. The generated macrocycle remains within the pocket while extending further outward.

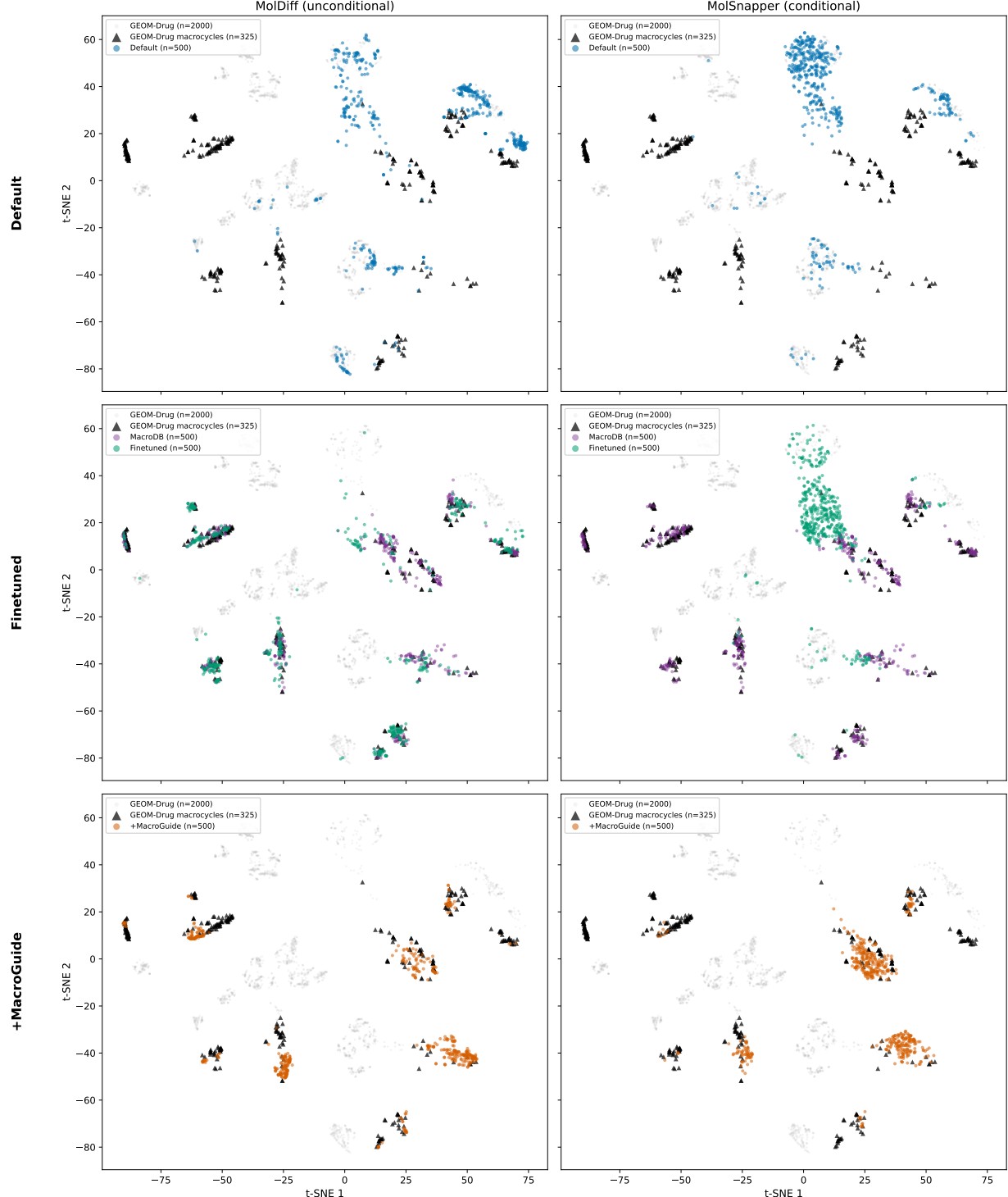

*Figure 26.* **Feature space comparison.** Topological features of generated molecules visualized alongside a subset of the training data using a shared t-SNE embedding. Macrocycles from GEOM-Drug are shown as triangles; in the finetuned setting, a subset of Macrocycle-DB is also included. In the unconditional setting (left), both finetuned and MACROGUIDE macrocycles align with the GEOM-Drug and MacroDB clusters. Under protein conditioning (right), the finetuned model's outputs collapse toward the default MolSnapper distribution, while MACROGUIDE preserves the manifold structure of the unconditional case.

## L. Integrating Bond-level Information: Challenges

The following strategies have been tested in order to integrate bond-level information in our macrocycle guidance function.

1. Maximizing the $H_1$ component **death edge probability** in our method.

2. Computing a **representative cycle** of our $H_1$ component. However this task is both computationally expensive and ambiguous, as a representative cycle is not unique.

3. Pre-selecting the **ordered set of atoms** to form the cycle (before sampling), and maximizing edge probabilities while minimizing chord probabilities.

4. Pre-selecting the **unordered set of atoms** to form the cycle (before sampling), and applying $H_1$ guidance on this subset during sampling based on distances computed from probabilities.

None of the attempts mentioned yield convincing results, which we explain with the following three reasons. First, MACROGUIDE already achieves a $99\%$ macrocycle rate without integrating bond information. Consequently one can hardly hope for significant improvements. Second, any minimization of a chord probability is challenged by the fact that minimizing the probability of a bond can be done both by pushing the atoms apart and by bringing them very close, without a clear way to choose which one. Third, any backpropagation of the bond-predictor model is much more computationally expensive than our current guidance, as well as relying on the quality and robustness of this model. As a consequence, we believe that trying further to integrate bond-level information is not a priority.

