# OpenReview forum: "MacroGuide: Topological Guidance for Macrocycle Generation"
_ICML.cc/2026/Conference — ICML 2026 regular_

### Official Review · Reviewer_TgFP · 2026-02-26

**Soundness:** 3
**Presentation:** 3
**Significance:** 3
**Originality:** 3
**Overall Recommendation:** 5
**Confidence:** 4

**Summary:**

this is a high-quality paper. It proposes a novel and elegant solution to an important and challenging problem in drug design – the ab initio generation of macrocycles. The paper’s structure is clear, writing concise, experiments robust and carefully designed. I would suggest accepting this paper.

**Compliance With Llm Reviewing Policy:**

Affirmed.

**Key Questions For Authors:**

no

**Limitations:**

yes

**Strengths And Weaknesses:**

Strengths:
1.	Critical Problem: Macrocycle discovery is an important field in drug design, but current generative models have difficulties generating them. This work clearly describes this gap, and points out the fundamental reason (data scarcity & hard-to-capture global topology).
2.	Novel, Elegant Solution:
a)	Novelty: MACROGUIDE implements “persistent homology” as the gradient guidance signal, and seamlessly incorporates it into the sampling process.
b)	Plug and Play: no training/finetuning needed, which greatly reduces the cost of deployment.
c)	Explainability: theorem 3.1 provides the theoretical foundation for the tuning of the cycle size, strengthening the credibility of the proposed method.
3.	Comprehensive, Robust Experiments:
a)	Sufficient Benchmarks: multiple benchmarks presented (no guidance, finetuned, naïve geometric guidance, torus noise initialization), together forms a comprehensive benchmark.
b)	Remarkable Improvements: MACROGUIDE increases macrocycle generation from 0~5% to nearly 100% (table 1 and 2), demonstrating its effectiveness.
c)	In-depth Quality Assessment: table 3 and 4 further assess the qualities of generated macrocycles, to prove that MACROGUIDE can not only “generate” macrocycles, but also “good” macrocycles.
d)	Section 4.2 shows that MACROGUIDE can also be used to generate bicyclic molecules; section 4.3 compares theoretical and empirical situations; the appendix further explores technological details and the robustness of MACROGUIDE, demonstrating authors’ profound understanding of the topic.
Weaknesses:
1.	Model Dependency: MACROGUIDE, a guidance framework, largely depends on the performances of its base model. While this work demonstrated that MACROGUIDE is effective, it does not explore whether MACROGUIDE can still function when the base model is subpar. The authors can include a simple experiment to verify this hypothesis.
2.	Definition of Macrocycles: the authors define macrocycles as “chordless cycles of size at least 12 atoms.” While this definition is clear, it is limited, and can leave out some meaningful macrocycles defined otherwise. The authors can elaborate on this in its work, paving paths for future works.

---

> ### Author Rebuttal · Authors · 2026-03-31
>
> We would like to sincerely thank the reviewer for their feedback and time they dedicated to our work. We greatly appreciate their recognition of our experimental robustness and the care taken in the study design. It is particularly encouraging to hear that the structure and clarity of our manuscript resonated with them, and we thank them for highlighting these strengths. We would like to address the points raised by the reviewer.
>
> **W1. Dependence on base model performance**
>
> To address the reviewer's insightful point regarding base model dependency, we evaluated MacroGuide’s robustness by simulating a degraded (subpar) base model. We injected increasing levels of Gaussian noise into the pre-trained weights of the MolDiff denoiser (relative to each parameter's standard deviation). As expected, base validity and connectivity steadily dropped as noise increased, collapsing entirely at 2% relative noise. MacroGuide cannot fix a fundamentally broken denoiser. However, **for all viable noise levels**, the degraded base model produced **fewer than 10%** macrocycles, whereas MacroGuide successfully steered more than **99%** of valid connected molecules into macrocycles. This demonstrates a key property of  our method: it decouples global topology from local chemistry. Macroguide robustly enforces macrocyclic topology, while relying on the base model only for local chemical correctness. The full quantitative results can be found in the Appendix.
>
> **W2. Alternative macrocycle definitions**
>
> We thank the reviewer for highlighting the potential ambiguity in the definition of a macrocycle. In our work, we define a macrocycle as a structure whose largest ring in the smallest set of smallest (SSSR) rings contains at least 12 atoms. This aligns with a common chemical definition and excludes cases such as anthracene, where the apparent outer 14-membered “cycle” is not chordless and hence is not considered a valid ring in the SSSR sense. Such structures also do not exhibit the characteristic beneficial properties associated with macrocycles, which motivates their exclusion.
>
> This definition allows for the presence of fused ring systems. As illustrated in Figure 2, macrocycles may contain multiple rings, but only the size of the relevant SSSR ring (the inner one in the intuitive sense) determines whether the molecule is classified as a macrocycle and contributes to the topological guidance.
>
> We note that alternative definitions exist, often differing in the minimum ring size threshold. Our framework can accommodate such variations by adjusting the target cycle size via the corresponding death-time parameter.
>
> Some works further restrict the definition of macrocycles to cyclic peptides, which captures a strict subset of our definition. MacroGuide is flexible enough to generate a broad spectrum of macrocycles, ranging from general structures to peptide-containing molecules, including cyclic peptides with non-canonical amino acids or non-standard cyclisation patterns. In practice, we observe that many generated molecules already contain peptide bonds. For applications specifically targeting cyclic peptides, one could combine the MacroGuide loss with models operating at the amino acid level to enforce cyclic structures, or alternatively use one of the existing cyclic-peptide-specific frameworks (see Appendix A.1).
>
> We remain open to alternative definitions of macrocycles and, should the reviewer deem it beneficial, we would be pleased to incorporate a dedicated discussion on these variations and their potential integration into our framework within the revised manuscript.
>
> We would like to sincerely thank the reviewer once again for the encouraging feedback and the insightful questions, which have helped us improve the rigor of our work. We hope that our responses have addressed all the points and concerns raised. We would be more than happy to provide further explanations or address any additional points the reviewer may raise.

---

> > ### Author Rebuttal · Reviewer_TgFP · 2026-04-05
> >
> > The authors have fully resolved my concerns.

---

> > > ### Author Response · Authors · 2026-04-07
> > >
> > > We are very grateful to the reviewer for the generous and encouraging feedback. We truly appreciate the time and care taken in engaging with our work. We are glad that our revisions were able to resolve the concerns and provide the necessary clarifications.

---

### Official Review · Reviewer_1iLi · 2026-03-09

**Soundness:** 3
**Presentation:** 3
**Significance:** 3
**Originality:** 3
**Overall Recommendation:** 4
**Confidence:** 4

**Summary:**

In this paper, the authors propose, Topological Guidance for Macrocycle Generation (MACROGUIDE), which uses persistent homology-based topological information to steer pretrained molecular generative models toward producing macrocycles. The model can be used in both unconditional settings and conditional ones involving protein pockets. Computationally, at each denoising step, ring formation is encouraged by persistent homology features. It has been found that the rate of macrocycle generation can be greatly increased at the same time, the quality metrics are maintained.

**Compliance With Llm Reviewing Policy:**

Affirmed.

**Key Questions For Authors:**

1. My major concern is the discussion of "bond" and graph representation of macrocycle in the paper. For instance, in Figure (3), the part for $F_{birth}^{H_1}$ seems to indicate that there is no "bond" there. This is incorrect! In fact, the cycle ring structure in macrocycle should be formed by covalent bonds. If there is a gap, this structure can not be called macrocycle, instead it should be called folded molecules. In the paper, the authors use the term "bond" many times without specifying covalent bonds and noncovalent bonds. In Figure (3), $F_{birth}^{H_1}$ should be the longest covalent bonds in the circle, while $F_{death}^{H_0}$ is most likely to be related to noncovalent bond. Note that, in general, noncovalent bonds are shorter than covalent bonds, as they are weaker in interaction.
2. The incorporation of $F_{birth}^{H_1}$ in guidance function seems to be weird. As explained above, this should be the covalent bond that has the largest length, i.e., the weakest covalent bond in the cycle. Why will this matters in the cycle forming?
3. VR complex is usually computational costly, why not using Alpha complex?

**Limitations:**

N.A.

**Strengths And Weaknesses:**

Strengths: The model incorporates topological information into generation process to significantly improve the generation rate without losing the quality.

Weakness: The discussion of the persistent homology-based score function/guidance function needs to be improved. Some of the illustration of topological features seems to be problematic.

---

> ### Author Rebuttal · Authors · 2026-03-31
>
> We sincerely thank the reviewer l for their time and careful evaluation of our work. We would like to address the points raised by the reviewer.
>
> **Q1, Q2. Incorporation of the $F^{H_1}_{birth}$ term**
>
> We agree with the reviewer that a macrocycle should be defined strictly through covalent bonds. Our method does not explicitly model non-covalent interactions, which we have now clarified in the manuscript. In both MolDiff and MolSnapper, molecules are generated with categorical bond types (single, double, triple, aromatic, or no bond), whose probabilities evolve during denoising. The final structure retains only the most probable bond. Bond strength is therefore captured only implicitly through bond type and interatomic distance.
>
> Although these models jointly represent atomic coordinates and bonds, the topological loss operates solely on coordinates. From a topological perspective, a configuration may exhibit cyclicity (i.e., a nontrivial $H_1$ feature) without forming a chemical ring – for instance when the corresponding atoms are too far apart to form covalent bonds. The $F^{H_1}_{birth}$ loss addresses this gap by constraining the longest edge in a candidate cycle to lie within a distance range consistent with typical covalent bonds.
>
> Importantly, the topological loss does *not* directly modify bond probabilities. Instead, by bringing atoms sufficiently close, it biases the denoiser toward bond formation. The $F^{H_1}_{\mathrm{birth}}$ threshold is set to 2 Å, slightly above typical covalent bond lengths (~1.4–1.8 Å), allowing flexibility during early denoising. If the longest edge in a cycle is below this value, it is no longer affected by the topological term and can only be refined by the denoiser. As shown in Figure 7, the longest edge typically remains near this threshold for much of the trajectory and is only shortened toward realistic bond lengths at later stages, driven entirely by the denoiser.
>
> Similarly, the $F^{H_1}_{\mathrm{death}}$ term does not model non-covalent bonds. Instead, this term focuses on bringing closer atom clusters that have drifted apart from each other. When the shortest distance between two clusters exceeds the 2 Å threshold, the relevant term has non-zero value and shortens the distance. And similarly to the birth term, the longest such distance stays close to 2 Å for most of the generation, and is shortened to a typical bond length towards the end (Figure 7).
>
> Empirically, this mechanism is effective: passing the macrocyclicity test requires a fully covalent cycle, and nearly all generated structures satisfy this criterion.
>
> Finally, in Figure 3, bonds are shown purely for visual clarity to indicate which interatomic distances are sufficiently short to plausibly correspond to bonds. However, this bond information is not passed to the topological guidance module. The figure illustrates how the $F^{H_1}_{birth}$​​ term constrains the longest edge in a candidate cycle so the generative model can subsequently realize it as a covalent bond. We appreciate the reviewer for pointing out this ambiguity and are improving the figure accordingly.
>
> **Q3. Vietoris-Rips versus Alpha filtration**
>
> We agree that Alpha complexes are computationally more efficient than Vietoris-Rips complexes. We considered them during development, but ultimately selected VR for three practical reasons:
>
> **1. Empirical performance:** Generative quality with Alpha complexes was strictly worse than VR. While VR maintained 99% chemical validity and a 100% macrocycle rate, Alpha guidance dropped validity to 85% and the macrocycle rate to 94%. Conceptually, VR applies a direct repulsive force across the shortest cross-ring chord, naturally keeping the macrocycle cavity open. Alpha optimizes a geometric circumradius, which can inadvertently distort local covalent bonds to satisfy the topological penalty.
>
> **2. Metric flexibility:** Alpha complexes strictly require Euclidean coordinates to construct the underlying Delaunay triangulation. VR only requires a generic pairwise distance matrix. This flexibility could be crucial for future work, as it leaves the door open to integrate non-Euclidean metrics (e.g., integrating bond-prediction logits) which Alpha cannot mathematically support.
>
> **3. Negligible runtime overhead:** Finally, while VR is theoretically more expensive, it is not the bottleneck of our pipeline. VR computation only accounts for ~8% of the sampling time, and our k-step guidance effectively reduces this to just ~2%, making the Alpha speed advantage negligible in practice.
>
> We thank the reviewer again for their thoughtful feedback. We hope these clarifications address the concerns raised by the reviewer. We are more than happy to answer any further questions.

---

> > ### Author Rebuttal · Reviewer_1iLi · 2026-04-05
> >
> > The authors have addressed my concerns very well. I have no further following up questions.

---

> > > ### Author Response · Authors · 2026-04-07
> > >
> > > We sincerely thank the reviewer for the careful reading of our rebuttal and for noting that the concerns have been addressed. We are very glad that our clarifications were helpful.
> > >
> > > In light of this, we would be most grateful if the reviewer might consider updating their assessment and score to reflect these improvements.

---

### Official Review · Reviewer_9RiF · 2026-03-09

**Soundness:** 2
**Presentation:** 3
**Significance:** 3
**Originality:** 3
**Overall Recommendation:** 4
**Confidence:** 4

**Summary:**

This paper proposes MACROGUIDE, a novel method for generating macrocycles. Macrocycles hold significant value in drug discovery, yet existing generative models struggle to effectively generate such structures. The core idea of MACROGUIDE is to utilize persistent homology, a topological tool, to add guidance signals during the denoising process of diffusion models. Specifically, at each denoising step, it constructs a Vietoris-Rips complex from the atomic point cloud and promotes macrocycle formation by optimizing topological features within it. This method does not require retraining the model and can be directly applied to existing pretrained diffusion models. Experimental results demonstrate that MACROGUIDE achieves good performance across metrics including chemical validity, molecular diversity, and structural rationality.

**Compliance With Llm Reviewing Policy:**

Affirmed.

**Final Justification:**

The rebuttal has partially addressed my concerns regarding empirical validation and methodological distinctions. I have raised my Originality rating from 2 to 3, as the authors more clearly articulated the novel combination of persistent homology with 3D diffusion guidance. However, my Soundness rating remains at 2, as some technical claims would benefit from additional experimental support in the revised manuscript. Overall, the work presents a practically useful solution to an important problem, and I maintain my recommendation of Weak Accept.

**Key Questions For Authors:**

Key Questions For Authors
1. Theorem 3.1 assumes molecules to be planar equilateral polygons, whereas actual macrocycles possess three-dimensional flexible conformations. Have you verified the correspondence between the number of atoms predicted by this formula and the actual number of atoms in generated macrocycles? If the prediction error is large, how does the method ensure generation of macrocycles with specified sizes?
2. If training iterations were increased or larger batch size were used, could the performance of the finetuning approach approach that of MACROGUIDE?
3. Section 4.1.2 attributes the performance collapse of the finetuned model to distribution shift, yet MACROGUIDE is likewise not trained on protein-conditioned data. What are the essential differences in feature space behavior between these two methods?

**Limitations:**

Limitations
The paper lacks adequate discussion of limitations. We suggest supplementing the following content:
- We suggest adding discussion of practical usage limitations, such as: persistent homology computation is sensitive to noise, how reliable is it in early diffusion steps, and whether guidance parameters need to be retuned for different molecular sizes.
- Dataset bias is not discussed: only 0.14% of GEOM-Drug are macrocycles, and the 40,496 molecules in Macrocycle-DB may be insufficient to cover diverse chemical space.

**Strengths And Weaknesses:**

Strengths
1. MACROGUIDE adopts a "plug-and-play" guidance strategy that enables macrocycle generation without retraining pretrained models, an advantage in practical scenarios with limited computational resources. Experiments show it only adds approximately 15% computational overhead, which can be further reduced through sparse guidance strategies.
2. Macrocycles possess unique pharmacological advantages but are difficult to generate; existing methods are mostly limited to cyclic peptides or require predefined scaffolds. The authors clearly distinguish the conceptual boundary between arbitrary macrocycles and cyclic peptides.

Weaknesses
1. Theorem 3.1 derives the relationship between ring size and death time based on the assumption of planar equilateral polygons, whereas actual molecules have three-dimensional flexible conformations. The paper does not validate the prediction accuracy of this linear approximation for real drug-like macrocycles. Additionally, the gradient masking strategy renders the guidance field non-conservative, lacking theoretical analysis of convergence properties for the diffusion process.
2. The finetuning baseline uses reduced batch size and limited training iterations, potentially not fully unleashing the method's potential.
3. The performance collapse of the finetuned model under conditional settings is attributed to distribution shift, but the paper does not explain why MACROGUIDE, which is likewise not trained on protein-conditioned data, remains unaffected. The absence of analysis on feature space distribution differences between the two methods.
4. The paper does not elaborate in depth on the unique advantages of MACROGUIDE's guidance mechanism. The application of persistent homology to molecular generation is not entirely novel, and the argument for methodological innovation is insufficient.

---

> ### Author Rebuttal · Authors · 2026-03-31
>
> We sincerely thank the reviewer for their time and thoughtful feedback on our work. We also greatly appreciate the positive feedback on its strong performance across metrics and low computational overhead.
>
> **W1, Q1. Ring size prediction accuracy**
>
> We agree that real macrocycles fold into complex 3D flexible conformations rather than remaining in idealized, planar crown shapes. That is precisely why we empirically validate this theoretical relationship. In Figure 5 we observe that the median sizes of the generated 3D macrocycles closely match the theoretical bounds for typical bond lengths (l = 1.0–1.5 Å). The slight underestimation by the formula is expected: due to the 3D folding effect, opposite sides come closer together, requiring slightly more atoms to reach the same distance d. Importantly, Figure 5 shows that **Theorem 3.1** provides a reliable monotonic baseline for controlling cycle size in practice.
>
> **W1. Gradient masking and theoretical convergence**
>
> We thank the reviewer for this insightful observation and agree that our guidance field is non-conservative. We have added a theoretical discussion on convergence in Appendix D. We highlight that non-conservative guidance does *not* preclude **convergence in distribution**. While a conservative drift guarantees convergence to the Boltzmann distribution, a non-conservative field still converges to a stationary distribution, provided that the conservative component prevents divergence. In this context, convergence to the marginal distribution is sufficient for achieving the desired generative performance.
>
> **W2, Q2. Finetuning performance**
>
> Finetuning was performed for a variable number of steps, with the final checkpoint chosen based on macrocycle generation rate. Further training did not improve the unconditional model and reduced protein-conditioned performance. The batch size was reduced due to larger molecules in the finetuning dataset, and GPU memory constraints. Nevertheless, finetuning remains a promising direction, potentially in combination with MacroGuide, and increasing availability of macrocyclic data will yield further performance gains.
>
> **W3, Q3. Feature space**
>
> The key difference lies in how each method handles topology under spatial constraints. MolSnapper is trained to approximate local geometric features (e.g., angle distribution) without explicit topology, which breaks down under protein constraints, reducing the macrocycle rate to **18%**. In contrast, MacroGuide directly optimizes topology via an independent gradient that enforces ring closure, maintaining a **99% macrocycle generation** rate even under these constraints. This distinction has been clarified in the manuscript.
>
> **W4. MacroGuide advantages**
>
> To our knowledge, MacroGuide is the first method enabling **direct 3D generation of arbitrary macrocycles**, a capability particularly relevant for drug discovery. It is also the first to allow direct optimization of molecular topological features via persistent homology, explicitly linking cyclicity with chemical structure.
> Related graph-based generation methods exist, some using persistent homology [1], but these operate on edge probabilities, and capture only the *number* of cycles - not their geometric size - and do not transfer easily to 3D. Topological features have also been applied in molecular classification [2,3] or combined with SMILES for VAE generation [4], but these methods do not allow direct topological optimization or 3D generation. We are happy to include additional related works if the reviewer has further specific pointers.
>
> **L1, L2. Further discussion**
>
> Persistent homology is robust to noise (see Bottleneck Stability Theorem). In early diffusion stages, high noise induces many random $H_1$ cycles. This is beneficial, as it spreads sparse topological gradients across atoms to form a coarse ring. As noise decays, the guidance refines these features. Empirically, Fig. 7 shows that topological features converge within the first 100 steps.
>
> Core MacroGuide parameters are chemically grounded and do *not* require retuning for different molecule sizes, provided the base model can handle the requested atom count. Desired macrocycle ring size can be adjusted with the target death time (**Theorem 3.1**).
>
> We agree that data scarcity and bias are important issues. In fact, the limited availability of macrocyclic data is a key motivation for our approach. Rather than relying on large datasets, MacroGuide enables flexible generation from general molecular distributions, allowing the model to explore a broader macrocyclic space.
>
> We thank the reviewer for the insightful feedback and for prompting these clarifications, which have been added to the manuscript. We hope our rebuttal addresses all key points and would appreciate reconsideration of the evaluation. We are very happy to answer any further questions.
>
> [1]arXiv:2512.19736
>
> [2]doi.org/10.1038/s41467-020-17035-5
>
> [3]arXiv:2311.10808
>
> [4]arXiv:2106.04464

---

> > ### Author Rebuttal · Reviewer_9RiF · 2026-04-04
> >
> > I appreciate the authors' detailed responses. While some concerns have been addressed, a few key issues remain that would benefit from further clarification or revision in the manuscript.
> > - The explanation that "MolSnapper breaks down under protein constraints" while MACROGUIDE maintains performance due to "direct topology optimization" is descriptive rather than analytical. I suggest adding a figure showing latent space visualizations or feature distributions comparing both methods under conditional generation, to empirically demonstrate the claimed behavioral difference.

---

> > > ### Author Response · Authors · 2026-04-07
> > >
> > > We thank the reviewer for the follow-up and for pushing us to provide a more analytical explanation. We conducted additional experiments to demonstrate the differences between finetuning and MacroGuide under protein conditioning, which are available at https://anonymous.4open.science/r/macro-guide/Rebuttal.pdf.
> > >
> > > Our analysis reveals that the performance gap stems from the **speed and robustness of the topological enforcement**: MacroGuide converges to macrocyclic topology within ~100 steps and resists perturbations, while finetuning encodes only a slow, fragile implicit bias that is easily overridden by protein-pocket constraints.
> > >
> > > **Observation: protein conditioning suppresses ring size.** Fig. 1 shows the distribution of the largest chordless ring size across methods. In the unconditional case, finetuning successfully shifts the ring size distribution above the macrocycle threshold of 12 atoms. Under protein conditioning, however, finetuning produces a large fraction of molecules with ring sizes 9-11, just below the threshold. The protein pocket's spatial and pharmacophore constraints thus act as a competing force that decreases cycle size. MacroGuide, by contrast, maintains a clean distribution concentrated at ring sizes 12-16 in both settings.
> > >
> > > **Analytical explanation: two competing forces.** The generation process under protein conditioning can be understood as balancing two objectives: (i) macrocyclicity (forming a large ring) and (ii) protein-pocket compatibility (satisfying pharmacophore constraints and avoiding steric clashes). These two forces compete: the pocket constraints favor compact geometries and specific atom placements, while macrocyclicity requires a globally extended ring.
> > >
> > > **For the finetuned model, macrocyclicity is encoded only as an implicit prior learned during finetuning**. Fig. 2 reveals that this prior is weak: the $H_1$ death feature (measuring ring size) drifts slowly and noisily toward the macrocyclic range over the entire 1000-step denoising trajectory. By contrast, MacroGuide's $H_1$ death converges into the target band within 100-200 steps. The weakness of the finetuned prior is further confirmed by perturbation experiments (Fig. 2-5): **after a mid-trajectory perturbation, MacroGuide recovers fully while the finetuned model does not**. In the protein-conditioned setting, the pocket constraints can be understood as a persistent perturbation applied throughout denoising, and since the finetuned model's implicit macrocyclicity bias is too weak and too slow to resist, the pocket-compatibility objective dominates, producing the near-miss ring sizes observed in Fig. 1.
> > >
> > > For MacroGuide, both macrocyclicity and pocket compatibility are explicitly encoded as concurrent objectives. The topological guidance locks in the macrocyclic ring structure within the first ~100-200 steps, and the remaining denoising steps can be spent refining the generated molecule. Because the macrocyclicity objective is enforced rapidly and robustly at every step, it cannot be overridden by the competing pocket forces. Importantly, MacroGuide does not forcefully fit a macrocycle where it does not belong: as shown in Fig. 7, the minimum distance to the pocket, the fraction of atoms within 4 Å of the pocket, and the steric clash rate all converge to virtually identical values across methods, indicating that **MacroGuide molecules maintain a tight fit without compromising pocket compatibility**. As shown in Fig. 8, the minimum distance to the deepest residue (GLU81) also remains the same, while the centroid distance increases, showing that MacroGuide macrocycles extend further outward from the pocket center to better utilize the available space and accommodate a larger ring. Fig. 9 visualizes this behavior for a representative molecule.
> > >
> > > **Feature space visualization.** A t-SNE embedding based on topological features ($H_0$ death times and $H_1$ birth/death, Fig. 6) confirms this analysis: in the unconditional setting, both methods produce macrocycles that align with the GEOM-Drug and Macrocycle-DB macrocycle clusters, but under protein conditioning, the finetuned model's outputs collapse toward the default MolSnapper distribution, while MacroGuide preserves the manifold structure of the unconditional case.
> > >
> > > In summary, the key advantage of MacroGuide is not merely that it optimizes topology directly, but that it does so *fast and robustly* (converging in ~100 steps and recovering from perturbations) whereas finetuning produces only a slow, fragile implicit bias that is easily overridden by protein-conditioning forces.
> > >
> > > We thank the reviewer again for their thoughtful questions, which helped us improve the work and provide additional insights that will be incorporated into the camera-ready manuscript version. We hope these clarifications have addressed all remaining concerns, and we would greatly appreciate it if the reviewer might take these improvements into account in their final assessment and score.

---

### Official Review · Reviewer_z72y · 2026-03-13

**Soundness:** 3
**Presentation:** 3
**Significance:** 3
**Originality:** 3
**Overall Recommendation:** 5
**Confidence:** 4

**Summary:**

This paper proposes MACROGUIDE, a diffusion guidance mechanism to steer pretrained small-molecule generative models toward the generation of macrocyclic molecules. Macrocycles are of great importance in drug discovery yet are also challenging. The proposed guidance provides topological control that encourages the generation of macrocycles, enables control over ring size, and can even promote the generation of bicyclic structures. The proposed approach is training-free, lightweight, general, and flexible, making it easy to integrate with existing molecular generative models.
In both unconditional generation and protein pocket–conditioned generation settings, the method significantly increases the proportion of generated macrocycles compared with finetuned models, and substantially outperforms naive approaches such as macrocycle-shaped noise initialization. In terms of molecular quality evaluation, including metrics related to 2D topology and 3D molecular conformation, the proposed method achieves higher PoseBusters (PB) pass rates than baseline models.
In general, the method is capable of generating chemically valid macrocyclic molecules, providing a promising direction for addressing the challenging task of macrocycle generation.

**Compliance With Llm Reviewing Policy:**

Affirmed.

**Final Justification:**

The authors provided extensive additional experiments in the rebuttal and addressed most of my previous concerns. I therefore am inclined to raise my score, as I believe this paper offers a strong technical contribution.

**Key Questions For Authors:**

Q1: In Tables 3 and 4, could the authors report the novelty with respect to the finetuning set only? Can the proposed guidance generate entirely new macrocyclic structures?

Q2: It is unclear whether MolSnapper was finetuned only on macrocyclic molecules without pocket conditioning. Would it be possible to perform pocket-conditioned finetuning (e.g., using docking software) to reduce the train-test mismatch?

Q3: Considering the well-established connections between flow matching velocity and diffusion scores, could the proposed guidance mechanism also be applied to flow matching models?

**Limitations:**

The experiments & evaluation are not strong enough. (see weakness)

**Strengths And Weaknesses:**

Soundness (Strength):
1. The use of persistent homology features (H1 birth/death and H0 death) as topological constraints for ring formation and molecular connectivity is mathematically well-motivated.
2. The paper establishes an empirical relationship between macrocycle size and the number of heavy atoms, which is supported by experimental validation.
3. The paper also discusses several practical issues, including gradient sparsity, efficiency, and potential extensions of the method, validating the practical effectiveness of the proposed method.

Soundness (Weakness):
1. The chosen base models are relatively weak: MolDiff (2023), MolSnapper (2024), making the improvement not convincing enough.
2. The pocket-conditioned generation is only evaluated on 1 pocket.
3. The paper reports several PB metrics, but it is unclear whether the evaluation corresponds to the full PB benchmark. The evaluation mainly focuses on geometric validity and topological properties, while more detailed assessments of 3D conformational/binding quality could further strengthen the evaluation, e.g., strain energy, relaxation RMSD, docking scores, docking RMSD.

Presentation:
The presentation is clear and easy to follow. Except that the paper does not clearly state in the main text that the pocket-conditioned experiment is conducted on only a single target.

Significance:
The research question is of great importance and challenging.

Originality:
PH guidance is novel.

---

> ### Author Rebuttal · Authors · 2026-03-31
>
> We thank the reviewer for their time and detailed review. We are truly grateful for the positive feedback on the mathematical motivation of MacroGuide and the clarity of its presentation. We now address the main points raised by the reviewer.
>
> **W1, Q3. Stronger base models and flow matching**
>
> To demonstrate the generality of MacroGuide, we apply it to FLOWR.root (Cremer et al. 2025), a state-of-the-art flow-matching model for molecules supporting unconditional and protein-conditioned generation, among other modes. We incorporate MacroGuide directly into the velocity field, preserving key advantages of flow-based methods such as fast sampling. Experiments are conducted on the BACE protein pocket using the provided setup.
>
> |  | FLOWR.root | +MacroGuide |
> |---|:---:|:---:|
> | Validity | 0.96 | 0.91 |
> | Macrocycle rate out of valid | 0.04 | 0.98 |
>
> We observe an impressive **20x increase** in valid macrocycle generation, demonstrating that MacroGuide is compatible with modern generative and flow-based frameworks, and can be applied reliably.
>
> **W2. Number of protein pockets**
>
> We extend the evaluation to 73 protein pockets from the CrossDocked dataset following the MolSnapper setup, and generate molecules using the original parameters without additional filtering.
>
> |  | MolSnapper | +MacroGuide |
> |---|:---:|:---:|
> | Validity (averaged over 73 pockets) | 0.67 | 0.74 |
> | Macrocycle rate out of valid | 0.05 | 0.98 |
>
> We observe a large increase in macrocycle generation alongside a modest improvement in validity. The baseline achieves >20% macrocycles in only **2 of 73 pockets**, primarily globular ones, and performs poorly on open pockets - precisely where macrocycles are most valuable for binding, which highlights a key limitation of the considered baseline.
>
> **W3. PoseBusters and other metrics**
>
> When reporting the overall PoseBusters pass rate, we considered all the metrics provided by PoseBusters. For clarity, we omitted some individual metrics from the tables, as they were either 100% or near 100% - for example, metrics related to molecular validity, since all macrocycles submitted to PoseBusters had already passed related validity checks, or metrics that assess issues that could not occur. We have updated the manuscript to clarify this.
>
> Strain energy is indirectly captured by the internal energy metric in PoseBusters, which measures the difference between the generated and relaxed molecular structures. We observe a significant improvement in this metric in the protein-conditioned case. We agree that additional metrics could further enrich the analysis; however, their performance is likely to depend more strongly on the underlying base model and its ability to accurately model protein-ligand interactions, rather than on the topological guidance itself.
>
> **Q1. Finetuned dataset novelty**
>
> This is a very interesting question, and we thank the reviewer for the suggestion. We have computed the metric, and in all cases it is equal to **1.0**. However, novelty is typically defined as a binary indicator of presence in the training set. In our setting, the finetuning dataset is relatively small and contains macrocycles of varying sizes, which limits the informativeness of this metric. More comprehensive alternatives of novelty include continuous measures such as fingerprint similarity, scaffold or 3D similarity, or comparisons to known binders targeting the protein of interest. The most appropriate choice of metric depends on the application, and we suggest considering these alternatives for a more nuanced notion of novelty.
>
> **Q2. Finetuning setup**
>
> MolSnapper has been finetuned on all data in Macrocycle-DB without using protein-ligand pairs, as the model incorporates protein constraints via additional conditional signals of pocket shape and pharmacophore features. This enables the same finetuning setup to be applied to both MolDiff and MolSnapper. We thank the reviewer for pointing out the lack of clarity here and have updated the manuscript accordingly.
>
> Models that explicitly learn from protein-ligand pairs can be finetuned on such data, which may help mitigate train-test mismatch. However, **only 3,780** spatially resolved pairs are available. Expanding this with docking-derived data is a potential direction, as much of Macrocycle-DB comes from binding screens where the target is known but the binding pose is not. As more macrocycle-protein pair data becomes available, finetuning on such pairs could yield increasingly strong results.
>
> **Q3. Flow matching**
>
> In W1 we show that our framework can indeed be readily **applied to flow-matching** formulations, expanding the range of base models suitable for macrocycle generation.
>
> We thank the reviewer for their careful consideration of our work and for the constructive feedback. We hope that the clarifications provided above help address the main concerns, and we kindly invite the reviewer to reconsider the evaluation in light of these points.

---

> > ### Author Rebuttal · Reviewer_z72y · 2026-04-03
> >
> > The authors provided extensive additional experiments in the rebuttal and addressed most of my previous concerns. I therefore am inclined to raise my score, as I believe this paper offers a strong technical contribution. That said, I still encourage the authors to include more binding-related evaluation in the final version, which would further strengthen the empirical support and practical relevance of the method.

---

> > > ### Author Response · Authors · 2026-04-04
> > >
> > > Dear Reviewer z72y,
> > >
> > > Thank you for your positive response and for recognizing the technical contribution of our work! We are thrilled that the additional experiments have adequately addressed your concerns, and we deeply appreciate your willingness to raise your score.
> > >
> > > We also highly value your suggestion regarding the inclusion of more binding-related evaluations to further strengthen the empirical support of our method. We will conduct a detailed analysis of the proposed metrics and include these evaluations in the final camera-ready version.
> > >
> > > Thank you again for your time and guidance in helping us improve our paper.

---

### Decision · Program_Chairs · 2026-04-30

**Decision:**

Accept (regular)

**Comment:**

This paper makes a clear and practically useful contribution by introducing a plug-and-play topological guidance mechanism for macrocycle generation, and the reviewers found both the idea and the empirical results strong. The rebuttal addressed most of the concerns about theory, baselines, and conditional evaluation, and the final consensus is clearly positive. Overall, this is a technically solid paper with convincing evidence and is above the ICML bar.